# Exploring The Role of Mean Teachers in Self-supervised Masked Auto-Encoders

[1,2]Youngwan Lee[*]  [2]Jeffrey Willette[*]  [1]Jonghee Kim  [2]Juho Lee  [2]Sung Ju Hwang

[1]Electronics and Telecommunications Research Institute (ETRI), South Korea
[2]Korea Advanced Institute of Science and Technology (KAIST), South Korea

## Abstract

Masked image modeling (MIM) has become a popular strategy for self-supervised learning (SSL) of visual representations with Vision Transformers. A representative MIM model, the masked auto-encoder (MAE), randomly masks a subset of image patches and reconstructs the masked patches given the unmasked patches. Concurrently, many recent works in self-supervised learning utilize the student/teacher paradigm which provides the student with an additional target based on the output of a teacher composed of an exponential moving average (EMA) of previous students. Although common, relatively little is known about the dynamics of the interaction between the student and teacher. Through analysis on a simple linear model, we find that the teacher conditionally removes previous gradient directions based on feature similarities which effectively acts as a *conditional momentum regularizer*. From this analysis, we present a *simple* SSL method, the Reconstruction-Consistent Masked Auto-Encoder (RC-MAE) by adding an EMA teacher to MAE. We find that RC-MAE converges faster and requires less memory usage than state-of-the-art self-distillation methods during pre-training, which may provide a way to enhance the practicality of prohibitively expensive self-supervised learning of Vision Transformer models. Additionally, we show that RC-MAE achieves more robustness and better performance compared to MAE on downstream tasks such as ImageNet-1K classification, object detection, and instance segmentation.

## 1 Introduction

The Transformer (Vaswani et al., 2017) is the *de facto* standard architecture in natural language processing (NLP), and has also surpassed state-of-the-art Convolutional Neural Network (He et al., 2016; Tan & Le, 2019) (CNN) feature extractors in vision tasks through models such as the Vision Transformer (Dosovitskiy et al., 2021) (ViT). Prior to the advent of ViTs, self-supervised learning (SSL) algorithms in the vision community (He et al., 2020; Chen et al., 2020c; Grill et al., 2020; Chen et al., 2021) utilized CNNs (*e.g.,* ResNet (He et al., 2016)) as a backbone, performing instance discrimination pretext tasks through contrastive learning (He et al., 2020; Chen et al., 2020c). Interestingly, self-distillation schemes (Grill et al., 2020; Caron et al., 2021) using a teacher consisting of an exponential moving average (EMA) of the previous students, (*i.e.*, a "mean" teacher) (Tarvainen & Valpola, 2017), have been shown to exhibit strong performance.

Inspired by the success of masked language modeling (MLM) pre-training in NLP, recent SSL approaches (Bao et al., 2022; Zhou et al., 2022; Xie et al., 2022; He et al., 2022; Assran et al., 2022) in the vision community have proposed forms of masked image modeling (MIM) pretext tasks, using ViT-based backbones. MIM is a simple pretext task which first randomly masks patches of an image, and then predicts the contents of the masked patches (*i.e.*, tokens) using various reconstruction targets, *e.g.*, visual tokens (Bao et al., 2022; Dong et al., 2021), semantic features (Zhou et al., 2022; Assran et al., 2022) and raw pixels (He et al., 2022; Xie et al., 2022). In particular, iBOT (Zhou et al., 2022) and MSN (Assran et al., 2022) use a self-distillation scheme for MIM by having the teacher network provide an encoded target (*i.e.*, feature representation) to match the encoded features from

---

[*]equal contribution

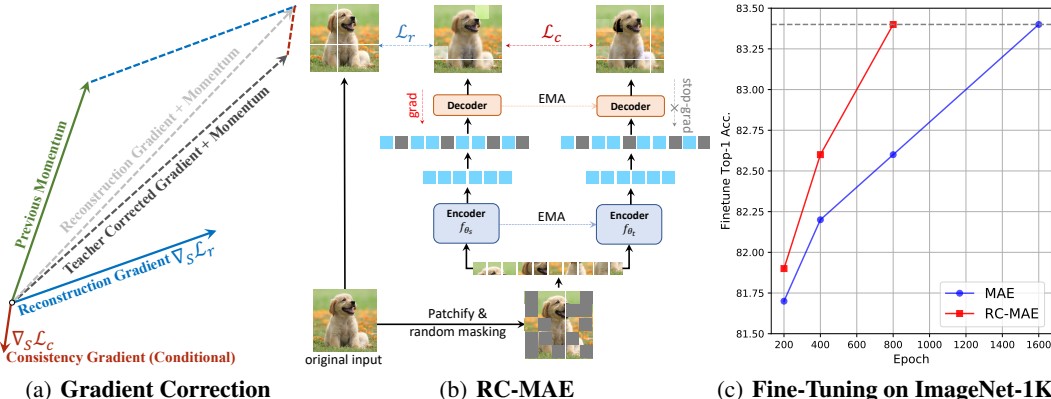

(a) **Gradient Correction**      (b) **RC-MAE**      (c) **Fine-Tuning on ImageNet-1K**

Figure 1: **Overview.** **(a)**: When the inputs which led to the **previous gradients** and **current gradients** are similar, the **consistency gradient** provides a conditional correction, allowing the student to learn from newer knowledge. **(b)**: In RC-MAE, the reconstructed patches from the student are compared with the original input (reconstruction loss $\mathcal{L_r}$), and with the predicted patches from the teacher (consistency loss $\mathcal{L_c}$). **(c)**: ImageNet-1K Fine-tuning top-1 accuracy curve: RC-MAE achieves comparable accuracy (83.4%) at 800 epochs compared to MAE trained for 1600 epochs.

the original image at a semantic feature level (*i.e.*, global abstraction). Methods using semantic-level target representations exhibit strong performance on image-level classification tasks. On the contrary, SimMIM (Xie et al., 2022) and MAE (He et al., 2022) provide pixel-level reconstructions of masked patches, and lead to superior performance on dense prediction tasks such as object detection and segmentation. However, self-distillation for pixel-level MIM has been under-explored as of yet.

A recent SSL approach, BYOL (Grill et al., 2020), has shown that a slight architectural asymmetry between a student and EMA teacher can create a stable model which outperforms previous contrastive learning methods. The success of BYOL (Grill et al., 2020) inspired empirical (Chen & He, 2021) and theoretical (Tian et al., 2021) analyses into what enables BYOL to effectively learn and avoid collapse with the EMA Teacher during pre-training. Still, despite the popularity of the EMA Teacher in SSL, relatively little is known about how the teacher interacts with the student throughout the training process.

In this work, we explore the dynamics of self-distillation in pixel-level MIM, *e.g.*, MAE. Through analyzing a simple linear model, we investigate the dynamics between the gradients of an image reconstruction loss and a teacher consistency loss, learning that the gradients provided by the teacher's consistency loss conditionally adjust the current reconstruction gradient by a weighted mixture of previous gradients. The weights of the mixture are derived from similarities between current and previous features. Thus, the teacher acts like a conditional momentum regularizer. For example, Fig. 1(a) shows the case where the inputs which created the previous gradient momentum are similar to the ones which created the current gradients. In this case, the teacher makes a conditional correction to remove the previous direction from the momentum, allowing the student to learn from the newer knowledge in the current batch. If however, the inputs which created both gradients are nearly orthogonal, the teacher would instead respond with minimal to no correction. We derive this conditional gradient effect in Proposition 4.1, and show evidence in both a simple linear model as well as in a deep ViT-based (Dosovitskiy et al., 2021) MAE model (Fig. 2).

To empirically validate our analysis of the contributions of EMA Teachers, we present a *simple* yet effective SSL approach, the *Reconstruction-Consistent Masked Auto-Encoder (RC-MAE)*, by equipping MAE with an EMA Teacher, and providing a consistency target. Additionally, we study the effects of using different image masking strategies between the student and teacher models on the consistency objective, finding that using the same mask generally leads to better performance in both pre-training and downstream tasks. The same mask tends to form an *orthogonal* objective (Fig. 3(b)) to the reconstruction loss, which has been shown (Suteu & Guo, 2019; Ajemian et al., 2013) to be beneficial for multi-task models as there is limited interference between tasks. This observation may be of interest to any future SSL works which leverage multiple pre-training objectives.

Our experiments follow the same architecture, settings, and pre-training recipe as MAE (He et al., 2022), and we find that the simple addition of a teacher (RC-MAE) consistently outperforms MAE in all model sizes (*e.g.*, ViT-S, ViT-B, and ViT-L) when fine-tuned for ImageNet classification. Additionally, we find that the teacher's conditional gradient correction we identified allows RC-MAE to converge faster compared to MAE (Fig. 1(c)), and RC-MAE outperforms recent self-distillation methods, MSN and iBOT, on dense prediction tasks such as object detection and instance segmentation. Furthermore, compared to recent self-distillation methods utilizing a mean teacher, RC-MAE realizes more efficiency in computation and memory due to the fact that both networks receive only a subset of patches instead of the whole image. **Our main contributions are as follows:**

1. We analyze the contribution of EMA Teachers in self-supervised learning, finding that the gradient provided by the teacher conditionally adjusts the current reconstruction gradient direction and magnitude conditioned on the similarity of current and previous features.

2. Using this knowledge, we propose a *simple*, yet effective approach for self-supervised pre-training of Vision Transformers, the *Reconstruction-Consistent Masked Auto-Encoder* (RC-MAE), which improves over vanilla MAE in terms of speed of convergence, adversarial robustness, and performance on classification, object detection, and instance segmentation tasks.

3. Thanks to its simplicity, RC-MAE achieves greater savings in both memory and computation compared to other state-of-the-art self-distillation-based MIM methods.

## 2 RELATED WORKS

In NLP, masked language modeling (MLM) is common for large-scale pre-training (Devlin et al., 2019; Radford et al., 2018) by predicting masked words. Similarly, ViT (Dosovitskiy et al., 2021; Liu et al., 2021; Lee et al., 2022) based masked image modeling (MIM) approaches (Zhou et al., 2022; Bao et al., 2022; He et al., 2022; Xie et al., 2022; Assran et al., 2022) for computer vision tasks have been proposed. These MIM approaches first apply a mask to patches of an image, and then the masked patches are predicted given the visible patches either at the token-level (Zhou et al., 2022; Bao et al., 2022; Assran et al., 2022) or pixel-level (Chen et al., 2020b; Xie et al., 2022; He et al., 2022). Token-level masked patch prediction (Zhou et al., 2022; Assran et al., 2022; Bao et al., 2022) *predicts* tokens or clusters of masked patches similar to MLM. Pixel-level prediction (Chen et al., 2020b; Xie et al., 2022; He et al., 2022) learns visual representations by *reconstructing* masked input patches at the RGB pixel-level.

Additionally, self-distillation (Grill et al., 2020; Caron et al., 2021; Chen et al., 2021) has been deployed in MIM methods by utilizing a teacher constructed from an exponential moving average (EMA-Teacher) of student weights, providing an additional target for the student. iBOT (Zhou et al., 2022) gives a full view of an image (*i.e.*, all patches) to the teacher network as an online tokenizer, offering a *token*-level target of the masked patches. Also giving a masked view to the student and a full view to the teacher, MSN (Assran et al., 2022) makes the output embeddings from an EMA-Teacher serve as a semantic feature *representation* target to the student. Likewise, BootMAE (Dong et al., 2022) also adopts an EMA-Teacher, providing a *feature*-level target to the student on top of the pixel-level MIM approach. A key difference from these self-distillation MIM approaches is that RC-MAE provides only unmasked patches to the teacher and student, instead of the full image. As a result, RC-MAE shows better **scalability** compared with recent methods (see. Table 6).

## 3 PRELIMINARIES

**The Masked Autoencoder.** (MAE) (He et al., 2022) is a self-supervised approach with a ViT encoder $f$ and decoder $h$, which randomly masks a portion of input patches, and then reconstructs the masked patches given the visible patches. Given an image $X \in \mathbb{R}^{C \times H \times W}$, MAE patchifies $X$ into $N$ non-overlapping patches $\tilde{X} \in \mathbb{R}^{N \times (P^2 \cdot C)}$ with a patch size of $P$ and randomly masks a subset of patches $\mathcal{M}$ (*i.e.*, mask tokens). The subset of visible patches $\mathcal{V}$ is input to the encoder to achieve latent representations: $z = f(\mathcal{V})$. Then, the decoder $h$ attempts to reconstruct $\mathcal{M}$ given the latent representations, $\hat{Y} = h(z; \mathcal{M})$, where $\hat{Y} \in \mathbb{R}^{N \times (P^2 \cdot C)}$ denotes the reconstructed patches. MAE

utilizes the mean-squared error reconstruction loss $\mathcal{L}_r$ which is only computed on masked patches:

$$\mathcal{L}_r = \frac{1}{|\mathcal{M}|} \sum_{i \in \mathcal{M}} \left\| \tilde{X}_i - \hat{Y}_i \right\|_2^2 \tag{1}$$

**The EMA Teacher.** The mean teacher model (Tarvainen & Valpola, 2017) is a temporal ensemble of previous student weights which provides an additional target to the student. Doing so has been shown to reduce the number of labels needed to achieve the same level of accuracy, and has become a core part of recent state-of-the-art SSL approaches as reviewed in Section 2. Predictions from the student and teacher are compared via a function such as mean squared error (Tarvainen & Valpola, 2017) or cross-entropy (Caron et al., 2021). Generally, the teacher $T$ is updated after every gradient step on the student $S$, using an exponential moving average of the student weights,

$$T^{(t)} = \alpha T^{(t-1)} + (1 - \alpha) S^{(t)} = \sum_{i=0}^{t} \alpha^i (1 - \alpha) S^{(t-i)}, \tag{2}$$

with a parameter $\alpha \in (0, 1)$. The additional target forms a ***consistency*** loss $\mathcal{L}_c$ between the teacher and the student predictions. Considering the mean squared error loss, and $\hat{Y}'$ being the prediction from the teacher model,

$$\mathcal{L}_c = \frac{1}{|\mathcal{M}|} \sum_{i \in \mathcal{M}} \left\| \hat{Y}_i - \hat{Y}_i' \right\|_2^2 \tag{3}$$

## 4    THE ROLE OF THE TEACHER

Although EMA teachers are common in recent SSL approaches, relatively little is known about the interaction between the student and teacher. Through analysis of a linear model that mirrors a MAE+Teacher objective, we will show how the gradients of both models interact. Considering a linear model for the student $S$ and teacher $T$ consisting of a single weight matrix, like an MAE, the objective is to reconstruct the original input $\mathbf{x}$ from a masked input $\tilde{\mathbf{x}} = \mathbf{x} \odot \mathbf{m}$, where $\odot$ is an elementwise multiplication and $\mathbf{m}$ is a random binary mask with a predefined masking ratio.

**Proposition 4.1.** *With the reconstruction and consistency objective (Eqs. (1) and (3)), the gradient contribution of the teacher ($\nabla_S \mathcal{L}_c$) adjusts the direction and magnitude of the reconstruction gradients ($\nabla_S \mathcal{L}_r$). The magnitude and direction of the adjustment from the teacher are conditional based on the similarity between the current and previous features. With $\hat{\mathbf{x}}$ representing an independent input from a previous timestep,*

$$\nabla_S \mathcal{L}_r + \nabla_S \mathcal{L}_c = \nabla_S \frac{1}{2} \| S\tilde{\mathbf{x}} - \mathbf{x} \|_2^2 + \nabla_S \frac{1}{2} \| S\tilde{\mathbf{x}} - StopGrad(T\tilde{\mathbf{x}}) \|_2^2$$

$$= S\tilde{\mathbf{x}}\tilde{\mathbf{x}}^\top - \mathbf{x}\tilde{\mathbf{x}}^\top + S\tilde{\mathbf{x}}\tilde{\mathbf{x}}^\top - T\tilde{\mathbf{x}}\tilde{\mathbf{x}}^\top$$

$$= \underbrace{S\tilde{\mathbf{x}}\tilde{\mathbf{x}}^\top - \mathbf{x}\tilde{\mathbf{x}}^\top}_{\nabla_S \mathcal{L}_r} - \underbrace{\left[ \sum_{i=1}^{t} \alpha^i \lambda \left[ \underbrace{S\hat{\mathbf{x}}\hat{\mathbf{x}}^\top - \mathbf{x}\hat{\mathbf{x}}^\top + S\hat{\mathbf{x}}\hat{\mathbf{x}}^\top - T\hat{\mathbf{x}}\hat{\mathbf{x}}^\top}_{\nabla_S \mathcal{L}_r + \nabla_S \mathcal{L}_c \text{ from } \hat{\mathbf{x}}} \right]^{(t-i)} \tilde{\mathbf{x}}\tilde{\mathbf{x}}^\top \right]}_{\nabla_S \mathcal{L}_c} \tag{4}$$

*Proof.* Please see Appendices B and B.1.1 ∎

The gradient of the consistency loss $\nabla_S \mathcal{L}_c$ is wholly represented by the last term on the RHS. Interestingly, there is a dot product $\hat{\mathbf{x}}^\top \tilde{\mathbf{x}}$ which gets distributed into every term of the sum. If we consider the dot product as cosine similarity $\cos(\hat{\mathbf{x}}, \tilde{\mathbf{x}}) = \hat{\mathbf{x}}^\top \tilde{\mathbf{x}} / \|\hat{\mathbf{x}}\| \|\tilde{\mathbf{x}}\|$, the possible cases for $\cos(\hat{\mathbf{x}}, \tilde{\mathbf{x}})$ for $\hat{\mathbf{x}}$ at a single arbitrary timestep $t$ are as follows $\cos(\hat{\mathbf{x}}, \tilde{\mathbf{x}}) \in \{-1, 0, 1, (0, 1), (-1, 0)\}$.

**Case 1: $\cos(\hat{\mathbf{x}}, \tilde{\mathbf{x}}) \in \{-1, 1\}$.** In this case, the resulting gradient from the last term on the RHS of Eq. (4) removes some amount of residual memory of the direction of a previous gradient. A cosine similarity of $|1|$ also means the inputs are collinear, and the gradient is invariant to the sign of $\hat{\mathbf{x}}^\top \tilde{\mathbf{x}}$.

**Case 2: $\cos(\hat{\mathbf{x}}, \tilde{\mathbf{x}}) = 0$.** In this case, There is zero contribution from the teacher for this term in the sum.

**Case 3: $|\cos(\hat{\mathbf{x}}, \tilde{\mathbf{x}})| \in (0, 1)$.** In this case, the component which contains the previous gradient will be weighted by the coefficient $\hat{\mathbf{x}}^\top \tilde{\mathbf{x}}$.

In all cases, due to the sign of the last term on the RHS of Eq. (4), the gradient of consistency loss conditionally removes residual memory of previous gradients. The magnitude of the removal is likewise conditional, which can be seen by using the triangle inequality to upper bound the final term of Proposition 4.1 to see that,

$$\|\nabla_S \mathcal{L}_c\| = \left\| \sum_{i=1}^{t} \alpha^i \lambda \nabla_{S^{(t-i)}} \mathcal{L}^{(t-i)} \tilde{\mathbf{x}} \tilde{\mathbf{x}}^\top \right\| \leq \sum_{i=1}^{t} \alpha^i \lambda \left\| \nabla_{S^{(t-i)}} [\ldots \hat{\mathbf{x}}^\top] \tilde{\mathbf{x}} \tilde{\mathbf{x}}^\top \right\| \quad (5)$$

leading to the conclusion that the magnitude and direction of the consistency gradient directly result from the similarities to features in the *recent* learning history, as $\alpha^i \in (0, 1)$ decays exponentially for distant timesteps in the past. The gradient of the student model can be bounded in a similar fashion, but without the decaying $\alpha$ coefficients which makes the bound much looser in general (see Appendix B.1).

**Empirical Test.** To test for this effect in a linear model, we conducted an experiment by training a linear model on data consisting of random samples $\mathbf{x} \in \mathbb{R}^{32}$ from random multivariate normal distributions (see Appendix E.1

Table 1: Input sequences used in Fig. 2

| Name | Input | Description |
|------|-------|-------------|
| Case 1 | $(\tilde{\mathbf{x}}, \tilde{\mathbf{x}})$ | same input twice (**same**) |
| Case 2 | $(\tilde{\mathbf{x}}, \hat{\mathbf{x}})$ | different inputs (**different**) |
| Case 3 | $(\tilde{\mathbf{x}}, \tilde{\mathbf{x}}')$ | same input with a different mask (**similar**) |

for further details). After each training iteration, we sampled an extra batch of data, and for each single point in the batch we constructed sequences consisting of two inputs described in Table 1. We then took a single gradient step and teacher update for the first input and calculated the gradient of the reconstruction and consistency loss on the second input.

**Expected Outcome.** Based on the similarity of the inputs, we would expect the consistency loss to produce a larger gradient for the same or similar inputs and a smaller gradient for different inputs. Additionally, the direction of the reconstruction and consistency gradient should be closer to opposite for case 1, and closer to orthogonal for case 2, with case 3 falling somewhere in-between. In Fig. 2 (top row), we in fact observe this trend, noting that the reconstruction loss produces a significantly larger gradient when the second gradient step is on different inputs due to the looser bound in Eq. (16).

**Interpretation.** This finding implies that the teacher plays a role of something akin to a gradient memory, where the teacher acts as a memory bank which retrieves the memory of recent gradients based on matching a query $\tilde{\mathbf{x}}$ to a key $\hat{\mathbf{x}}$ in the memory bank. For novel inputs which do not match anything in recent memory, the teacher responds with a minimal correction, letting the student learn more from the reconstruction signal. If the query and key match, however, the teachers gradient will conditionally remove some directional information contained in the previous gradient. This allows the student to move in a direction which favors new knowledge gained from the current input, and cancels out some previous momentum. This process is illustrated in Fig. 1(a). In Appendix D, we show that the same terms appear in the context of a deep model, with the dot product appearing at each semantic feature level. However, in a complex model with nonlinearities, the resulting gradient direction becomes harder to interpret. Even so, in Fig. 2 (bottom row), we empirically find the same underlying trend in the gradient norms and directions when analyzing RC-MAE (a ViT based model).

## 5    RECONSTRUCTION-CONSISTENT MASKED AUTO-ENCODER

In this section, we utilize the analysis from Section 4, and present a *simple* self-supervised approach, *Reconstruction-Consistent Masked Auto-Encoder (RC-MAE)*. The overview of RC-MAE is illustrated in Fig. 1(b). RC-MAE is a ViT-based version of the simple linear model outlined in Section 4 where the total objective consists of a reconstruction and consistency loss from the original image and a mean teacher, respectively. The teacher network shares the same architecture as the student, consisting of an encoder $f_{\theta_t}$ (*e.g.*, ViT (Dosovitskiy et al., 2021)) and a decoder $h_{\theta_t}$.

At timestep $t$, the model parameters of the teacher $\theta_t$ are updated to be the exponential moving average (EMA) of the student model parameters $\theta_s$ (Eq. (2)). While BYOL (Grill et al., 2020), DINO (Caron et al., 2021), and MSN (Assran et al., 2022) make the student network mimic the teachers semantic-level representation, for RC-MAE, the consistency and reconstruction targets are both pixel-level representations. We would expect that the two pixel-level objectives would lead to better performance on dense prediction tasks. Additionally, we would expect the conditional

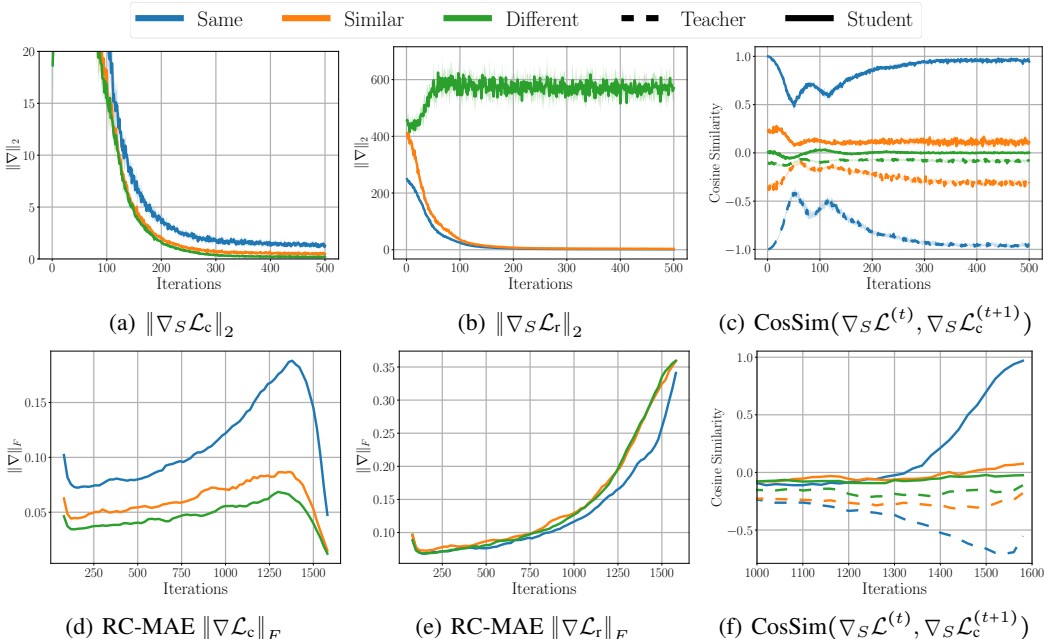

Figure 2: *Top Row:* Linear model. *Bottom Row:* RC-MAE (a ViT). Both models show the same trends. For three input sequences in Table 1, we performed one gradient step/teacher update and then calculated $\nabla_S \mathcal{L}_r$, and $\nabla_S \mathcal{L}_c$ at the next step. *Figs. 2(a) and 2(d)*: $\|\nabla_S \mathcal{L}_c\|_F$ is larger when $\cos(\hat{\mathbf{x}}, \tilde{\mathbf{x}})$ is large (Eq. (5)). *Figs. 2(b) and 2(e)*: The **same** and **different** lines do not follow the same order for the reconstruction loss. *Figs. 2(c) and 2(f)*: The direction of $\nabla_S \mathcal{L}_c$ is conditioned on $\cos(\hat{\mathbf{x}}, \tilde{\mathbf{x}})$. *All together:* For $\nabla_S \mathcal{L}_c$, when $\cos(\hat{\mathbf{x}}, \tilde{\mathbf{x}}) = 1$ there tends to be larger move in a negative direction while $\cos(\hat{\mathbf{x}}, \tilde{\mathbf{x}}) \approx 0$ there tends to be a smaller move in an approximately orthogonal direction.

momentum corrections from the consistency gradient to allow for quicker and more stable rate of convergence, due to our interpretation in Section 4 of the momentum corrections supplied by the teacher.

To investigate masking strategies for the teacher, we define two types of RC-MAE, `RC-MAE-S` and `RC-MAE-D`. `RC-MAE-S` uses the **s**ame mask tokens for both student and teacher networks. Conversely, `RC-MAE-D`, uses **d**ifferent mask tokens for each network. For both variants, given an image, we randomly sample respective mask tokens $\mathcal{M}$ with a mask ratio (*e.g.*, 75% like MAE). The visible patches are then given to the student and teacher networks, which reconstruct all patches. With the consistency loss Eq. (3), the RC-MAE objective is:

$$\frac{1}{|\mathcal{M}|} \sum_{i \in \mathcal{M}} (\underbrace{\left\| \tilde{X}_i - \hat{Y}_i \right\|^2}_{\text{reconstruction}} + \underbrace{\left\| \hat{Y}_i - \hat{Y}'_i \right\|^2}_{\text{consistency}}), \tag{6}$$

where $i$ is the token index and $\hat{Y}, \hat{Y}'$ denote the reconstructed patches from the decoders of the student and teacher networks, respectively. For the reconstruction target, we use standard normalized patch pixels as done with MAE (He et al., 2022). For `RC-MAE-D` the student and teacher process different visible patches, and the total loss is only calculated on the student's mask token locations $\mathcal{M}_s$. Unless otherwise specified, RC-MAE means RC-MAE-S in the rest of this paper.

## 6 EXPERIMENTS

To validate the effectiveness of the EMA Teacher in RC-MAE, we compare RC-MAE to an otherwise identical MAE (He et al., 2022) baseline. For the EMA Teacher, We use ViT (Dosovitskiy et al., 2021) as an encoder and transformer decoder with the same depth and dimension as MAE. For a fair comparison with MAE, we follow the same implementation as MAE's official code. For experiments, we pre-train on ImageNet-1K and evaluate linear probing (LN) and end-to-end fine-tuning (FT) for classification and COCO object detection & instance segmentation for which we

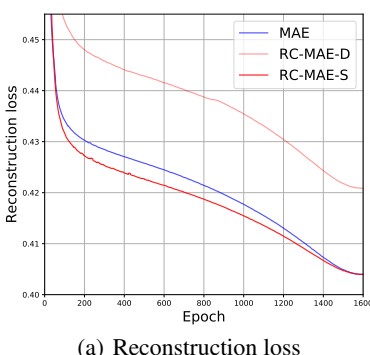 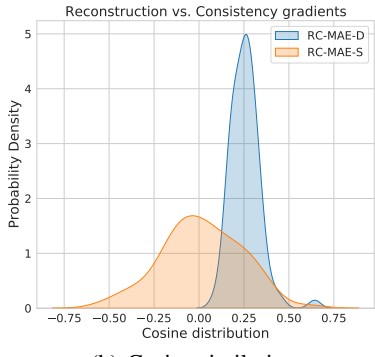

|  |  |
|:---:|:---:|
| (a) Reconstruction loss | (b) Cosine similarity |

Figure 3: **(a)**: $\mathcal{L}_r$ is high for RC-MAE-D compared with MAE, even though RC-MAE-D outperforms MAE in downstream tasks (see Table 2). **(b)**: RC-MAE-D likely causes a conflict between $\mathcal{L}_r$ and $\mathcal{L}_c$ (described in Appendix C), and the resulting $\cos(\nabla\mathcal{L}_r, \nabla\mathcal{L}_c)$ shows a positive bias. Interestingly, RC-MAE-S shows a similar distribution as the special cases studied in Fig. 2(f) indicating that the same situations arise in an i.i.d. training setting.

Table 2: **Different mask types for the teacher network in RC-MAE.** The 'Same' means that the teacher network uses the same mask tokens as the student and 'Diff.' is the opposite. FT and LN denote end-to-end fine-tuning and linear probing on ImageNet-1K, respectively.

| Method | Mask | FT | LN | AP$^{box}$ | AP$^{mask}$ |
|---|---|---|---|---|---|
| MAE | - | 83.4 | 67.3 | 50.3 | 44.9 |
| **RC-MAE-D** | **D**iff. | **83.6** | **68.7** | 50.7 | 45.2 |
| **RC-MAE-S** | **S**ame | **83.6** | 68.4 | **51.0** | **45.4** |

Table 3: **Different masking ratio.** GPU time means pre-training time (hours) on 8 V100-32GB GPUs environment.

| Method | Epoch | Mask Ratio | FT | GPU Time |
|---|---|---|---|---|
| MAE | 800 | 75% | 83.2 | 125.5h |
| **RC-MAE** | 800 | 75% | 83.4 | 166.6h |
| **RC-MAE** | 800 | 80% | 83.4 | 142.6h |
| **RC-MAE** | 800 | 85% | 83.2 | 136.4h |
| MAE | 1600 | 75% | 83.4 | 256.5h |
| **RC-MAE** | 1600 | 75% | **83.6** | 331.1h |
| **RC-MAE** | 1600 | 80% | 83.5 | 283.3h |

use the Mask R-CNN benchmark (Li et al., 2021) for dense prediction. Further details regarding implementation and training settings can be found in Appendix E.

## 6.1 ABLATION STUDY

**Mask token for Teacher: Same vs. Different.** To investigate the effect of masking strategies between teacher and student, we evaluate RC-MAE variants against a plain MAE. Results are shown in Table 2. Both RC-MAE-S and RC-MAE-D outperform MAE, achieving a higher accuracy gap in linear probing (1.4% / 1.1%) compared to fine-tuning (0.2%). Notably, RC-MAE-S shows higher performance than RC-MAE-D on object detection and instance segmentation while the performance gap for classification slightly favors RC-MAE-D. To analyze the differences between RC-MAE-D and RC-MAE-S, we recorded the cosine similarity between the reconstruction and consistency gradients throughout training Fig. 3(b). We found that RC-MAE-D resulted in a cosine similarity with a positive bias, while RC-MAE-S appears to be centered near zero. We hypothesize that a conflict between the pixel-level reconstruction and consistency objectives causes the biased cosine similarity in RC-MAE-D, leading to lower pre-training and subsequent fine-tuning performance on these tasks. The base conflict likely stems from the fact that MAE models tend to give lower-quality reconstructions on visible patches. Therefore, when a patch is masked in the student but not the teacher, the teacher provides a low-quality consistency target, interfering with the high-quality original input reconstruction target.

**Masking ratio & Training epoch.** We compare to MAE using different masking ratios, measuring pre-training time on the same 8-GPU machine. Per-iteration, RC-MAE is slower due to the added teacher network. However, compared to MAE with 75% mask & 1600 epochs (default), RC-MAE with 80% mask & 800 epochs achieves the same level of performance while using 55% less time. Fig. 1(c) illustrates fine-tuning accuracy curves of RC-MAE (75% mask & 800 epochs) and MAE (75% mask & 1600 epochs). It is easy to observe that RC-MAE reaches higher accuracy quicker than MAE. As the teacher is the only difference between the models, the increased rate of convergence is a direct result from the conditional teacher gradients.

Table 4: **End-to-end fine-tuning on ImageNet-1K.** All results (top-1 acc.) are trained with 224 × 224 input. MAE (our impl.) were trained using the official code with the training configuration on the same 8 V100-32GB GPUs environment by ourselves. Note that there are reproducing issues.

| Method | Pre-Train Data | Pre-Train Epochs | ViT-S | ViT-B | ViT-L |
|---|---|---|---|---|---|
| Supervised. (Touvron et al., 2021) | IN1K w/ labels | 300 | 79.9 | 81.8 | 82.6 |
| DINO (Caron et al., 2021) | IN1K | 800 | 81.5 | 82.8 | - |
| MoCo v3 (Chen et al., 2021) | IN1K | 300 | 81.4 | 83.2 | 84.1 |
| BEiT (Bao et al., 2022) | IN1K+DALLE | 800 | 81.7 | 83.2 | 85.2 |
| MSN (Assran et al., 2022) | IN1K | 600 | - | 83.4 | - |
| iBOT (Zhou et al., 2022) | IN1k | 800 | **82.3** | 84.0 | 84.8 |
| BootMAE (Dong et al., 2022) | IN1k | 800 | - | **84.2** | 85.9 |
| MAE (He et al., 2022) | IN1K | 1600 | - | 83.6 | 85.9 |
| MAE (our impl.) | IN1K | 1600 | 81.8 | 83.4 | 85.5 |
| **RC-MAE** | IN1K | 1600 | 82.0 | 83.6 | **86.2** |

Table 5: **COCO object detection and segmentation** using Mask R-CNN with ViT-Base backbone. For fair comparison, we follow the benchmarking transfer learning protocol (Li et al., 2021).

| Method | Pre-Train Data | $AP^{box}$ | $AP^{mask}$ |
|---|---|---|---|
| Superivsed (He et al., 2022) | IN1K w/ labels | 47.9 | 42.9 |
| MoCo v3 (Chen et al., 2021) | IN1K | 47.9 | 42.7 |
| BEiT (Bao et al., 2022) | IN1K+DALLE | 49.8 | 44.4 |
| MSN (Assran et al., 2022) | IN1K | 46.6 | 41.5 |
| iBOT (Zhou et al., 2022) | IN1K | 47.3 | 42.2 |
| MAE (He et al., 2022) | IN1K | 50.3 | 44.9 |
| **RC-MAE** | IN1K | **51.0** | **45.4** |

Table 6: **Resource Comparison** with Self-distillation based MIM methods: **GPU Memory** and **runtime** measured using 8 V100-32GB GPUs with a batch size of 128 for ViT-L.

| Method | Memory | Throughput |
|---|---|---|
| MAE | 84G | 533 imgs/s |
| MSN | 183G | 78 imgs/s |
| iBOT | 227G | 123 imgs/s |
| BootMAE | 98G | 376 imgs/s |
| **RC-MAE** | **95G** | **441 imgs/s** |

## 6.2 END-TO-END FINE-TUNING ON IMAGENET-1K

Table 4 shows ImageNet fine-tuning results. Note that our reproduced MAE (*e.g.*, ViT-B and ViT-L) on the same 8 GPUs machine shows slightly lower accuracy than the original MAE due to a known reproduction issue. We assume that the slight difference in accuracy stems from the different GPU environments. For all ViT backbone sizes, RC-MAE consistently outperforms the reproduced MAE. While ViT-S and ViT-B achieve lower accuracy than iBOT or BootMAE, RC-MAE with ViT-L surpasses the state-of-the-art methods. These results suggest that the reconstruction-consistent scheme could have stronger benefits for large-scale models.

## 6.3 OBJECT DETECTION AND SEGMENTATION ON COCO

To validate the pixel-level representation quality of RC-MAE, we analyze performance on object detection and instance segmentation on COCO (Lin et al., 2014). Following the same training protocol (Li et al., 2021) and implementation details as MAE (He et al., 2022), we fine-tune Mask R-CNN (He et al., 2017) with a ViT-Base backbone pre-trained by RC-MAE for 100 epochs. The training details are described in Appendix E.4. Table 5 summarizes the detection and segmentation results. RC-MAE improves over the MAE baseline by 0.7% box AP and 0.5% mask AP. We note that both RC-MAE and MAE outperform iBOT by large margins, even though iBOT shows higher ImageNet classification accuracy. These results suggest that the pure pixel-level objectives allow models to learn richer representations at the region or pixel-level. Moreover, as illustrated in Fig. 4, while iBOT and MSN attend to uncorrelated regions given the query patches, RC-MAE and MAE can focus more sharply on the pertinent regions near the query patches. These results indicate that pixel-level MIM is more advantageous than semantic-level MIM for dense prediction tasks.

## 6.4 RESOURCE COMPARISON: MEMORY AND COMPUTATION TIME

In terms of memory consumption and computation cost, we conduct a resource comparison with the self-distillation-based MIM methods, *e.g.*, MSN, iBOT, and BootMAE as shown in Table 6. More details about how to measure are described in Appendix F.2. Since adding the EMA Teacher to MAE, RC-MAE shows more memory usage and runtime. However, compared with state-of-the-art self-distillation methods, RC-MAE requires less memory usage and computation cost. This result

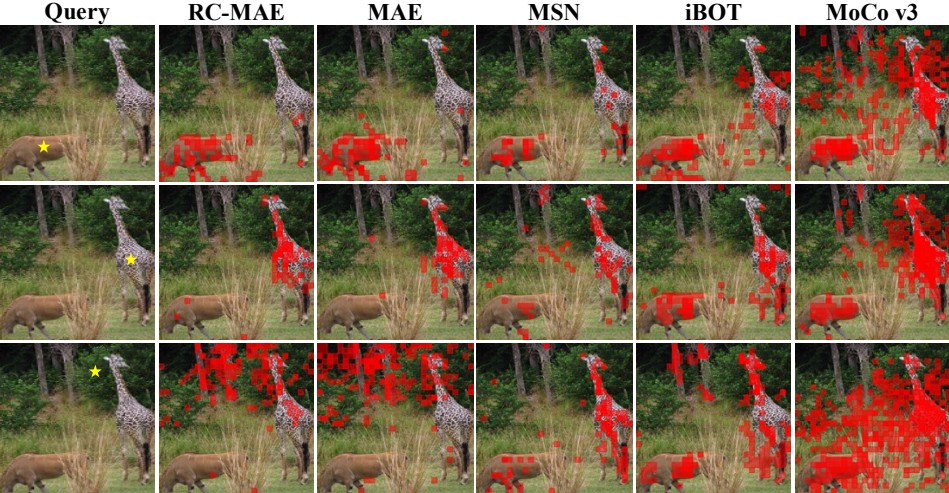

Figure 4: **Attention maps obtained by thresholding 10% mass of the query patch (★)**. RC-MAE & MAE attend regions more precisely with respect to the query patch while MSN, iBOT, and MoCo v3 are likely to attend the unrelated region with the query patch.

Table 7: **Robustness Evaluation** on ImageNet-variants:ImageNet-C (Hendrycks & Dietterich, 2019), ImageNet-A (Hendrycks et al., 2021b), ImageNet-R (Hendrycks et al., 2021a), and ImageNet-Sketch (Wang et al., 2019). Except for ImageNet-C which is measured in terms of mean Corruption Error (mCE), top-1 accuracy is used as the metric for IN-A,-R, and -Sketch.

| Method | IN-C (mCE ↓) | IN-A (top-1 ↑) | IN-R (top-1 ↑) | IN-Sketch (top-1 ↑) |
|---|---|---|---|---|
| MAE | 51.7 | 35.9 | 48.3 | 34.5 |
| **RC-MAE** | **49.9** | **36.5** | **49.5** | **35.8** |

can be attributed to the fact that other methods feed the full image to the teacher network, whereas RC-MAE only delivers visible patches (no mask patches) to the teacher same as the student.

## 6.5    ROBUSTNESS EVALUATION ON IMAGENET

We evaluate the robustness of models on four ImageNet variants: ImageNet-C (Hendrycks & Dietterich, 2019),-A (Hendrycks et al., 2021b),-R (Hendrycks et al., 2021a), and -Sketch (Wang et al., 2019), which are common benchmarks for evaluating robustness to perturbations. Table 7 shows the robustness comparison with MAE using the ViT-B backbone. RC-MAE outperforms MAE consistently on all robustness datasets, indicating that the conditional gradient corrections from the teacher result in a more robust solution.

## 7    CONCLUSION

In this work, we have provided an analysis and empirical verification of the nature of contributions from EMA teachers in self-supervised learning. We have found that teachers provide a conditional momentum correction, where the direction and magnitude is conditioned on current and previous feature similarities. Equipped with our analysis, we proposed a *simple* and effective self-supervised learning approach, *RC-MAE*, combining a masked auto-encoder (MAE) with an EMA teacher. In our experiments, we observed that the teacher enables quicker convergence of RC-MAE which achieves a gain in ImageNet classification with increased robustness. Furthermore, RC-MAE outperforms recent self-supervised models on dense prediction tasks. Since our aim is to analyze the dynamics of the EMA teacher, we design a *simple* approach to validate our findings, which results in better computation and memory efficiency compared to other self-distillation methods. For future work, we look forward to seeing further applications of our analysis. For example, could the conditional corrections from the teacher be adequately incorporated into an Adam-like optimizer? If so this would remove the need to have a full model copy as a teacher and lead to cost savings and faster convergence in a wide variety of settings outside of self-supervised learning.

## 8    ACKNOWLEDGEMENT

This work was partly supported by Institute of Information & Communications Technology Planning & Evaluation(IITP) grant funded by the Korea government(MSIT) (No. 2014-3-00123, Development of High Performance Visual BigData Discovery Platform for Large-Scale Realtime Data Analysis, No. 2020-0-00004, Development of Previsional Intelligence based on Long-term Visual Memory Network), (No.2022-0-00124, Development of Artificial Intelligence Technology for Self-Improving Competency-Aware Learning Capabilities), (No. RS-2022-00187238, Development of Large Korean Language Model Technology for Efficient Pre-training), and (No.2019-0-00075, Artificial Intelligence Graduate School Program(KAIST)).

## 9    REPRODUCIBILITY STATEMENT

In this section, we guide where the sections are for reproducibility.

- Linear model experiments outlined in Section 4 → Appendix E.1
- ImageNet experiments such as pre-training, fine-tuning, linear probing, and robustness evaluation → Appendix E.3
- COCO object detection and instance segmentation → Appendix E.4

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

## A  APPENDIX

We will briefly describe the contents of each section of this appendix below:

- Appendix B: Detailed Derivations of Proposition 4.1 the Main Text.
- Appendix C: An Intuitive Explanation of Differences Between RC-MAE-D and RC-MAE-S.
- Appendix D Does the Linear Model Extend to Deep Models?
- Appendix E: Extra Experimental Details.
- Appendix F: Additional Experiments Not Covered in the Main Text.

## B  LINEAR MODEL DERIVATION

With the weights $S$ and $T$ signifying the student and teacher weights (if no explicit timestep superscript $(t)$ is given, then the current timestep is assumed), $\tilde{\mathbf{x}}$ and $\mathbf{x}$ representing the masked and unmasked inputs, and StopGrad(.) signifying an operation which stops the propagation of the gradients, $\lambda$ as the learning rate, and considering the following assumptions,

**Assumption B.1.** *Both the masked input $\tilde{\mathbf{x}}$ and $\mathbf{x}$ have zero mean.*

**Assumption B.2.** *For all geometric sums, we assume the sequence $N$ is long enough so that $\sum_{i=0}^{N} \alpha^i (1 - \alpha) \approx 1$.*

The gradient of loss w.r.t the full linear MAE+Teacher model is,

$$\nabla_S \mathcal{L}(S, T, \tilde{\mathbf{x}}, \mathbf{x}) = \nabla_S \mathcal{L}_r(S, \tilde{\mathbf{x}}, \mathbf{x}) + \nabla_S \mathcal{L}_c(S, T, \tilde{\mathbf{x}}, \mathbf{x}) \tag{7a}$$

$$= \nabla_S \frac{1}{2} \| S\tilde{\mathbf{x}} - \mathbf{x} \|_2^2 + \nabla_S \frac{1}{2} \| S\tilde{\mathbf{x}} - \text{StopGrad}(T\tilde{\mathbf{x}}) \|_2^2 \tag{7b}$$

$$= \underbrace{S\tilde{\mathbf{x}}\tilde{\mathbf{x}}^\top - \mathbf{x}\tilde{\mathbf{x}}^\top}_{\nabla_S \mathcal{L}_r} + \underbrace{(S - T)\tilde{\mathbf{x}}\tilde{\mathbf{x}}^\top}_{\nabla_S \mathcal{L}_c} \tag{7c}$$

**Remark B.1.** *In essence, the MAE objective learns the transformation from square diagonal blocks to off-diagonal blocks of a feature covariance matrix.*

Setting the gradient of the reconstruction loss (which is the plain MAE objective) equal to zero, re-arranging and taking the expectation w.r.t the joint data and masking distribution $p(\mathbf{x}, \tilde{\mathbf{x}})$ we can see,

$$0 = S\tilde{\mathbf{x}}\tilde{\mathbf{x}}^\top - \mathbf{x}\tilde{\mathbf{x}}^\top \tag{8a}$$

$$\mathbf{x}\tilde{\mathbf{x}}^\top = S\tilde{\mathbf{x}}\tilde{\mathbf{x}}^\top \tag{8b}$$

$$\mathbb{E}_{p(\mathbf{x},\tilde{\mathbf{x}})}\big[\mathbf{x}\tilde{\mathbf{x}}^\top\big] = \mathbb{E}_{p(\mathbf{x},\tilde{\mathbf{x}})}\big[S\tilde{\mathbf{x}}\tilde{\mathbf{x}}^\top\big] \tag{8c}$$

$$\mathbb{E}_{p(\mathbf{x},\tilde{\mathbf{x}})}\big[\mathbf{x}\tilde{\mathbf{x}}^\top\big] = S \iint_{\mathbf{x},\tilde{\mathbf{x}}} p(\mathbf{x},\tilde{\mathbf{x})}\tilde{\mathbf{x}}\tilde{\mathbf{x}}^\top d\mathbf{x} d\tilde{\mathbf{x}} \tag{8d}$$

$$\mathbb{E}_{p(\mathbf{x},\tilde{\mathbf{x}})}\big[\mathbf{x}\tilde{\mathbf{x}}^\top\big] = S\mathbb{E}_{p(\tilde{\mathbf{x}})}\big[\tilde{\mathbf{x}}\tilde{\mathbf{x}}^\top\big] \tag{8e}$$

$$\Sigma_{\mathbf{x},\tilde{\mathbf{x}}} = S\Sigma_{\tilde{\mathbf{x}},\tilde{\mathbf{x}}} \tag{8f}$$

$$\tag{8g}$$

This means that the optimal $S$ represents the transformation which takes the square positive semi-definite covariance matrix $\Sigma_{\tilde{\mathbf{x}},\tilde{\mathbf{x}}}$ and transforms it into another off diagonal block of the covariance matrix $\Sigma_{\mathbf{x},\tilde{\mathbf{x}}}$, which is a part of some larger block covariance matrix,

$$\begin{bmatrix} \Sigma_{\mathbf{x},\mathbf{x}} & \Sigma_{\mathbf{x},\tilde{\mathbf{x}}} \\ \Sigma_{\tilde{\mathbf{x}},\mathbf{x}} & \Sigma_{\tilde{\mathbf{x}},\tilde{\mathbf{x}}} \end{bmatrix}$$

## B.1 What is The Role of The Teacher?

Although the student-teacher paradigm with an EMA teacher is common Tarvainen & Valpola (2017); Grill et al. (2020); Assran et al. (2022); Zhou et al. (2022); Caron et al. (2021); Bao et al. (2022), it is still not well understood how the teacher objective directly interacts with and helps the student. In addition to the explanations in Section 4, here we will provide a complete derivation of Proposition 4.1.

### B.1.1 Proof of Proposition 4.1

The gradient contribution from the teacher and the consistency loss conditionally remove previous gradient directions based on similarities of current and previous features. Using the linear model derivation, the full gradient of an MAE+Teacher is given by,

*Proof.*

$$\nabla_S \mathcal{L} = \nabla_S \mathcal{L}_r + \nabla_S \mathcal{L}_c \tag{9a}$$

$$= \nabla_S \frac{1}{2} \|S\tilde{\mathbf{x}} - \mathbf{x}\|_2^2 + \nabla_S \frac{1}{2} \|S\tilde{\mathbf{x}} - \text{StopGrad}(T\tilde{\mathbf{x}})\|_2^2 \tag{9b}$$

$$= S\tilde{\mathbf{x}}\tilde{\mathbf{x}}^\top - \mathbf{x}\tilde{\mathbf{x}}^\top + S\tilde{\mathbf{x}}\tilde{\mathbf{x}}^\top - T\tilde{\mathbf{x}}\tilde{\mathbf{x}}^\top \tag{9c}$$

$$= 2S\tilde{\mathbf{x}}\tilde{\mathbf{x}}^\top - \mathbf{x}\tilde{\mathbf{x}}^\top - \left[ (1-\alpha)S + \alpha(1-\alpha)S^{(t-1)} + \cdots + \alpha^t(1-\alpha)S^{(t-t)} \right]\tilde{\mathbf{x}}\tilde{\mathbf{x}}^\top \tag{9d}$$

$$= 2S\tilde{\mathbf{x}}\tilde{\mathbf{x}}^\top - \mathbf{x}\tilde{\mathbf{x}}^\top - \left[ (1-\alpha)S + \sum_{i=1}^{t} \alpha^i(1-\alpha)S^{(t-i)} \right]\tilde{\mathbf{x}}\tilde{\mathbf{x}}^\top \tag{9e}$$

$$= 2S\tilde{\mathbf{x}}\tilde{\mathbf{x}}^\top - \mathbf{x}\tilde{\mathbf{x}}^\top - \left[ (1-\alpha)S + \sum_{i=1}^{t} \alpha^i(1-\alpha)\left[ S + \left[ \lambda \sum_{j=t-i}^{t-1} \nabla_{S^{(j)}} \mathcal{L}^{(j)} \right] \right] \right]\tilde{\mathbf{x}}\tilde{\mathbf{x}}^\top \tag{9f}$$

$$= 2S\tilde{\mathbf{x}}\tilde{\mathbf{x}}^\top - \mathbf{x}\tilde{\mathbf{x}}^\top - \left[ S + \sum_{i=1}^{t} \alpha^i(1-\alpha)\left[ \lambda \sum_{j=t-i}^{t-1} \nabla_{S^{(j)}} \mathcal{L}^{(j)} \right] \right]\tilde{\mathbf{x}}\tilde{\mathbf{x}}^\top \tag{9g}$$

$$= 2S\tilde{\mathbf{x}}\tilde{\mathbf{x}}^\top - \mathbf{x}\tilde{\mathbf{x}}^\top - S\tilde{\mathbf{x}}\tilde{\mathbf{x}}^\top - \left[ \sum_{i=1}^{t} \alpha^i(1-\alpha)\left[ \lambda \sum_{j=t-i}^{t-1} \nabla_{S^{(j)}} \mathcal{L}^{(j)} \right] \right]\tilde{\mathbf{x}}\tilde{\mathbf{x}}^\top \tag{9h}$$

$$= \underbrace{S\tilde{\mathbf{x}}\tilde{\mathbf{x}}^\top - \mathbf{x}\tilde{\mathbf{x}}^\top}_{\nabla_S \mathcal{L}_r} - \underbrace{\left[ \sum_{i=1}^{t} \alpha^i(1-\alpha)\left[ \lambda \sum_{j=t-i}^{t-1} \nabla_{S^{(j)}} \mathcal{L}^{(j)} \right] \right]\tilde{\mathbf{x}}\tilde{\mathbf{x}}^\top}_{\nabla_S \mathcal{L}_c} \tag{9i}$$

where Eq. (9f) is a result of the fact that $S^{(t-1)}$ can be stated in relation to $S$ by re-arranging $S^{(t-1)} - \lambda\nabla_{S^{(t-1)}}\mathcal{L}^{(t-1)} = S$, and further decomposition of $S^{(t-1)}$ in the same way can yield any desired term in the sequence until reaching $S^{(0)}$. For a more concise expression, the indices of the sums can be re-arranged like so,

$$\nabla_S \mathcal{L} = \nabla_S \mathcal{L}_r + \nabla_S \mathcal{L}_c \tag{10a}$$

$$= S\tilde{\mathbf{x}}\tilde{\mathbf{x}}^\top - \mathbf{x}\tilde{\mathbf{x}}^\top - \lambda \left[ \sum_{i=1}^{t} \alpha^i (1-\alpha) \left[ \sum_{j=t-i}^{t-1} \nabla_{S^{(j)}} \mathcal{L}^{(j)} \right] \right] \tilde{\mathbf{x}}\tilde{\mathbf{x}}^\top \tag{10b}$$

$$= S\tilde{\mathbf{x}}\tilde{\mathbf{x}}^\top - \mathbf{x}\tilde{\mathbf{x}}^\top - \lambda \left[ \sum_{i=1}^{t} \sum_{j=t-i}^{t-1} \alpha^i (1-\alpha) \nabla_{S^{(j)}} \mathcal{L}^{(j)} \right] \tilde{\mathbf{x}}\tilde{\mathbf{x}}^\top \tag{10c}$$

$$= S\tilde{\mathbf{x}}\tilde{\mathbf{x}}^\top - \mathbf{x}\tilde{\mathbf{x}}^\top - \lambda \left[ \sum_{j=0}^{t-1} \sum_{i=0}^{j} \alpha^{t-i} (1-\alpha) \nabla_{S^{(j)}} \mathcal{L}^{(j)} \right] \tilde{\mathbf{x}}\tilde{\mathbf{x}}^\top \tag{10d}$$

$$= S\tilde{\mathbf{x}}\tilde{\mathbf{x}}^\top - \mathbf{x}\tilde{\mathbf{x}}^\top - \lambda \left[ \sum_{j=0}^{t-1} \nabla_{S^{(j)}} \mathcal{L}^{(j)} \sum_{i=0}^{j} \alpha^{t-i} (1-\alpha) \right] \tilde{\mathbf{x}}\tilde{\mathbf{x}}^\top \tag{10e}$$

$$= S\tilde{\mathbf{x}}\tilde{\mathbf{x}}^\top - \mathbf{x}\tilde{\mathbf{x}}^\top - \lambda \left[ \sum_{j=0}^{t-1} \nabla_{S^{(j)}} \mathcal{L}^{(j)} \alpha^{(t-j)} - \alpha^{t+1} \right] \tilde{\mathbf{x}}\tilde{\mathbf{x}}^\top \tag{10f}$$

$$\approx S\tilde{\mathbf{x}}\tilde{\mathbf{x}}^\top - \mathbf{x}\tilde{\mathbf{x}}^\top - \lambda \left[ \sum_{j=0}^{t-1} \alpha^{(t-j)} \nabla_{S^{(j)}} \mathcal{L}^{(j)} - 0 \right] \tilde{\mathbf{x}}\tilde{\mathbf{x}}^\top \tag{10g}$$

$$= \underbrace{S\tilde{\mathbf{x}}\tilde{\mathbf{x}}^\top - \mathbf{x}\tilde{\mathbf{x}}^\top}_{\nabla_S \mathcal{L}_r} - \underbrace{\lambda \left[ \sum_{i=1}^{t} \alpha^i \nabla_{S^{(t-i)}} \mathcal{L}^{(t-i)} \right] \tilde{\mathbf{x}}\tilde{\mathbf{x}}^\top}_{\nabla_S \mathcal{L}_c} \tag{10h}$$

where Eq. (10g) uses the fact that the full geometric sum converges to 1 and therefore $\alpha^{(t+1)} \approx 0$. We can then expand the gradient term to include the full gradient at the previous timesteps,

$$\nabla_S \mathcal{L}_r + \nabla_S \mathcal{L}_c = S\tilde{\mathbf{x}}\tilde{\mathbf{x}}^\top - \mathbf{x}\tilde{\mathbf{x}}^\top - \lambda \left[ \sum_{i=1}^{t} \alpha^i \nabla_{S^{(t-i)}} \mathcal{L}^{(t-i)} \right] \tilde{\mathbf{x}}\tilde{\mathbf{x}}^\top \tag{11a}$$

$$= \underbrace{S\tilde{\mathbf{x}}\tilde{\mathbf{x}}^\top - \mathbf{x}\tilde{\mathbf{x}}^\top}_{\nabla_S \mathcal{L}_r} - \underbrace{\lambda \left[ \sum_{i=1}^{t} \alpha^i \left[ S\hat{\mathbf{x}}\hat{\mathbf{x}}^\top - \mathbf{x}\hat{\mathbf{x}}^\top + (S-T)\hat{\mathbf{x}}\hat{\mathbf{x}}^\top \right]^{(t-i)} \right] \tilde{\mathbf{x}}\tilde{\mathbf{x}}^\top}_{\nabla_S \mathcal{L}_r} \tag{11b}$$

With $\hat{\mathbf{x}}$ signifying a different i.i.d. sample from the dataset which was present in the previous batch. Therefore, by the last term on the RHS, the gradient of the consistency loss is composed of residual effects from the previous gradient steps on different inputs which will have different overall effects based on the result of $\hat{\mathbf{x}}\hat{\mathbf{x}}^\top \tilde{\mathbf{x}}\tilde{\mathbf{x}}^\top$ and $\mathbf{x}\hat{\mathbf{x}}^\top \tilde{\mathbf{x}}\tilde{\mathbf{x}}^\top$. Consider the cosine similarity $\cos(\hat{\mathbf{x}}, \tilde{\mathbf{x}}) = \hat{\mathbf{x}}^\top \tilde{\mathbf{x}} / \|\hat{\mathbf{x}}\| \|\tilde{\mathbf{x}}\|$, the possible cases for $\cos(\hat{\mathbf{x}}, \tilde{\mathbf{x}})$ are as follows $\cos(\hat{\mathbf{x}}, \tilde{\mathbf{x}}) \in \{-1, 0, 1, (0, 1), (-1, 0)\}$.

**Case 1: $\cos(\hat{\mathbf{x}}, \tilde{\mathbf{x}}) \in \{-1, 1\}$.** For this case, a cosine similarity of $|1|$ means the points are collinear, existing in the same line, which makes the gradient invariant to the sign of $\hat{\mathbf{x}}^\top \tilde{\mathbf{x}}$. The removal of the previous gradient direction will have the largest impact in this case.

**Case 2: $\cos(\hat{\mathbf{x}}, \tilde{\mathbf{x}}) = 0$.** For this case, $\hat{\mathbf{x}}^\top \tilde{\mathbf{x}} = 0$, which zeros out the whole term. There is no removal of the previous gradient direction for this term in the recursion.

**Case 3: $|\cos(\hat{\mathbf{x}}, \tilde{\mathbf{x}})| \in (0, 1)$.** In this case, the component which contains the previous gradient will be weighted by the coefficient $\zeta = \hat{\mathbf{x}}^\top \tilde{\mathbf{x}}$.

$$\nabla_S \mathcal{L}_r + \nabla_S \mathcal{L}_c = S\tilde{\mathbf{x}}\tilde{\mathbf{x}}^\top - \mathbf{x}\tilde{\mathbf{x}}^\top - \left[ \sum_{i=1}^{t} \alpha^i \zeta_i \lambda \left[ S\hat{\mathbf{x}} - \mathbf{x} + (S-T)\hat{\mathbf{x}} \right]^{(t-i)} \right] \tilde{\mathbf{x}}^\top \tag{12a}$$

A generic linear model with a generic weight parameter $\Theta$ which has a gradient in the form of $\frac{\partial \mathcal{L}}{\partial \Theta} = (\Theta \tilde{\mathbf{x}} - \mathbf{x}) \tilde{\mathbf{x}}^\top$, can be interpreted as a gradient direction and magnitude for every component in

the vector $(\Theta\tilde{\mathbf{x}} - \mathbf{x})$ with a further conditional direction and magnitude projected to the components of $\Theta$ by the final $\tilde{\mathbf{x}}^\top$ term. Both the reconstruction and consistency losses in Eq. (7) follow this form. Therefore, in each case above, the correction provided to S by the consistency loss is weighted conditionally based on the dot product $\hat{\mathbf{x}}^\top\tilde{\mathbf{x}}$, and the final projection of the current feature $\tilde{\mathbf{x}}^\top$. $\qquad\square$

**Remark B.2.** *1 The upper bound on the gradient of the consistency loss can be bounded with the triangle inequality.*

We can examine the following term which uses the previous result from Eq. (10),

$$\nabla_S \mathcal{L}_c = -\left[\sum_{i=1}^{t} \alpha^i \lambda \nabla_{S^{(t-i)}} \mathcal{L}^{(t-i)}\right]\tilde{\mathbf{x}}\tilde{\mathbf{x}}^\top \tag{13a}$$

distributing the the outer terms and using the triangle inequality, we can see that,

$$\|\nabla_S \mathcal{L}_c\| = \left\|\left[\sum_{i=1}^{t} \alpha^i \lambda \nabla_{S^{(t-i)}} \mathcal{L}^{(t-i)}\right]\tilde{\mathbf{x}}\tilde{\mathbf{x}}^\top\right\| \le \sum_{i=1}^{t} \alpha^i \lambda \left\|\nabla_{S^{(t-i)}} \mathcal{L}^{(t-i)}\tilde{\mathbf{x}}\tilde{\mathbf{x}}^\top\right\| \tag{14a}$$

The most significant (judged by $\alpha$ coefficients) term in the upper bound of the norm of the consistency loss is when $i = 1$. Therefore the norm of the consistency loss from the teacher is likely to decrease if the current input is not similar to the previous input. Likewise, the most significant term in the upper bound increases when the current input has a high similarity to the input $\hat{\mathbf{x}}^{(t-1)}$. One should also note that the upper bound of the reconstruction loss can be decomposed via a similar line of reasoning,

$$G(\mathcal{L}_r) = S\tilde{\mathbf{x}}\tilde{\mathbf{x}}^\top - \mathbf{x}\tilde{\mathbf{x}}^\top \tag{15a}$$

$$= \left(S^{(t-1)} - \lambda\nabla_{S^{(t-1)}} \mathcal{L}^{(t-1)}\right)\tilde{\mathbf{x}}\tilde{\mathbf{x}}^\top - \mathbf{x}\tilde{\mathbf{x}}^\top \tag{15b}$$

$$= \left(S^{(0)} - \lambda\sum_{i=1}^{t}\nabla_{S^{(t-i)}} \mathcal{L}^{(t-i)}\right)\tilde{\mathbf{x}}\tilde{\mathbf{x}}^\top - \mathbf{x}\tilde{\mathbf{x}}^\top \tag{15c}$$

Focusing only on the non constant terms, we can see that,

$$\left\|\sum_{i=1}^{t}(\lambda\nabla_{S^{(t-i)}} \mathcal{L}^{(t-i)})\tilde{\mathbf{x}}\tilde{\mathbf{x}}^\top\right\| \le \sum_{i=1}^{t} \lambda \left\|\nabla_{S^{(t-i)}} \mathcal{L}^{(t-i)})\tilde{\mathbf{x}}\tilde{\mathbf{x}}^\top\right\| \tag{16}$$

It is easy to see there are no exponentially decreasing coefficients on the terms for the previous time steps, making the upper bound for the reconstruction loss larger than that of the consistency loss. This should lead to the reconstruction loss having a larger gradient contribution for novel inputs when compared to the consistency loss, as empirically shown in Fig. 2.

## C  WHY DOES RC-MAE-S OUTPERFORM RC-MAE-D

RC-MAE-S uses the same mask between the teacher and the student, while RC-MAE-D uses a different mask between the teacher and the student. Empirically, we observed a difference in cosine similarity between the gradients of $\mathcal{L}_r$ and $\mathcal{L}_c$ for the different masking strategies in the consistency loss as depicted in (Fig. 3). In general, a different masking strategy $\tilde{\mathbf{x}}$ and $\hat{\mathbf{x}}$ forces $S\tilde{\mathbf{x}}$ towards the prediction of $T\hat{\mathbf{x}}$ but there is no reason to expect that this objective aligns well with the original reconstruction target in practice.

For example, MAE models are known to give poor reconstructions on masked patches, because the masked patches do not receive any gradient signal during the pre-training phase. Therefore, on patches which are unmasked in the student and masked in the teacher, RC-MAE-D provides a consistency objective which interferes with the reconstruction objective by giving a noisy target of lower quality, which is the likely cause which leads to worse pre-training performance as shown in Fig. 3 and also lower finetuning performance as shown in Table 2.

## D  DO THE FINDINGS FROM THE LINEAR MODEL EXTEND TO DEEP MODELS?

Although we have analyzed a simple linear model, a more complex deep model with non-linearities can be broken down in a similar way. To see this with a simple network consisting of a single hidden layer $B\sigma(A\tilde{\mathbf{x}})$ with an elementwise non-linearity $\sigma$, and a quadratic consistency loss

$\mathcal{L}_{\mathrm{c}} = \frac{1}{2} \| B\sigma(A\tilde{\mathbf{x}}) - D\sigma(C\tilde{\mathbf{x}}) \|_2^2$, where the teacher weights are $\{C, D\}$, and the student weights are $\{A, B\}$. The teacher at a single layer can be expanded according to,

$$D\sigma(C\tilde{\mathbf{x}}) = D\sigma\left(\left[(1-\alpha)A + \alpha C^{(t-1)}\right]\tilde{\mathbf{x}}\right) \tag{17a}$$

$$= D\sigma\left(\left[(1-\alpha)A + \sum_{i=1}^{t} \alpha^i (1-\alpha)A^{(t-i)}\right]\tilde{\mathbf{x}}\right) \tag{17b}$$

$$= D\sigma\left(\left[(1-\alpha)A + \sum_{i=1}^{t}\left[\alpha^i(1-\alpha)\left(A + \lambda\sum_{j=t-i}^{t-1}\nabla_{A^{(i)}}\mathcal{L}^{(j)}\right)\right]\right]\tilde{\mathbf{x}}\right) \tag{17c}$$

$$= D\sigma\left(\left[A + \sum_{i=1}^{t}\left[\alpha^i(1-\alpha)\left(\lambda\sum_{j=t-i}^{t-1}\nabla_{A^{(j)}}\mathcal{L}^{(j)}\right)\right]\right]\tilde{\mathbf{x}}\right) \tag{17d}$$

$$= D\sigma\left(\left[A + \sum_{i=1}^{t} \alpha^i \lambda \nabla_{A^{t-i}}\mathcal{L}^{(t-i)}\right]\tilde{\mathbf{x}}\right) \tag{17e}$$

$$= D\sigma\left(\left[A + \sum_{i=1}^{t} \alpha^i \lambda [\dots \hat{\mathbf{x}}^\top]^{(t-i)}\right]\tilde{\mathbf{x}}\right) \tag{17f}$$

where the fourth step is a result of rearranging the sums following the same steps as Eq. (10). The term $[\dots \hat{\mathbf{x}}^\top]$ results from the fact that we know the previous gradient ends with the term $\hat{\mathbf{x}}^\top$, where $\hat{\mathbf{x}}$ is another i.i.d. sample from a previous batch, due to the chain rule applied as follows,

$$\frac{\partial \mathcal{L}_c}{\partial A} = \frac{\partial \mathcal{L}}{\partial B\sigma(A\tilde{\mathbf{x}})}\frac{\partial B\sigma(A\tilde{\mathbf{x}})}{\partial \sigma(A\tilde{\mathbf{x}})}\frac{\partial \sigma(A\tilde{\mathbf{x}})}{\partial A\tilde{\mathbf{x}}}\frac{\partial A\tilde{\mathbf{x}}}{\partial A} \tag{18a}$$

$$= \left[(B\sigma(A\tilde{\mathbf{x}}) - D\sigma(C\tilde{\mathbf{x}}))^\top B \operatorname{diag}(\sigma'(A\tilde{\mathbf{x}}))\right]^\top \tilde{\mathbf{x}}^\top \tag{18b}$$

$$= \left[\left(B\sigma(A\tilde{\mathbf{x}}) - D\sigma\left(\left[A + \sum_{i=1}^{t}\alpha^i\lambda[\dots\hat{\mathbf{x}}^\top]^{(t-i)}\right]\tilde{\mathbf{x}}\right)\right)^\top B \operatorname{diag}(\sigma'(A\tilde{\mathbf{x}}))\right]^\top \tilde{\mathbf{x}}^\top \tag{18c}$$

This shows that although intermediate representations are more complicated, they contain similar terms, and the dot product term is present at every intermediate representation. Having this term in each semantic feature space is useful, as the dot product in the high dimensional pixel space may carry less meaning. Empirically, in Fig. 2 we observed the same overall effects in the deep ViT based RC-MAE (bottom row) as we did in the simple linear model (top row).

# E  EXPERIMENT DETAILS

## E.1  LINEAR MODEL

To perform the linear model experiment outlined in Section 4, we first generated a dataset consisting of 10 random Gaussian clusters. Each cluster has a randomly sampled mean vector $\boldsymbol{\mu}_i \sim U(-3, 3)$, and each $\boldsymbol{\mu}_i \in \mathbb{R}^{32}$. With each mean vector, we also sample a corresponding covariance matrix $\Sigma_i \sim \text{Wishart}(V_i)$ with 54 degrees of freedom (1.5 × dimensionality). $V_i$ is a diagonal matrix with the diagonal being sampled from $U(.25, .35)$. We then sample 200 datapoints from each of the 10 Gaussians, resulting in a dataset with 1000 total instances. Further experimental settings can be found in Table 8. An example of the dataset in two dimensions with four clusters can be seen in Fig. 5.

The student and teacher $S, T \in \mathbb{R}^{32 \times 32}$ consist of a single weight matrix with a bias parameter. We randomly mask the components of the input vector according to the masking ratio. The goal of the linear model is then to reconstruct the full input given the masked input with a reconstruction $\mathcal{L}_{\mathrm{r}}$ and consistency $\mathcal{L}_{\mathrm{c}}$ loss according to the linear objectives in Eq. (7).

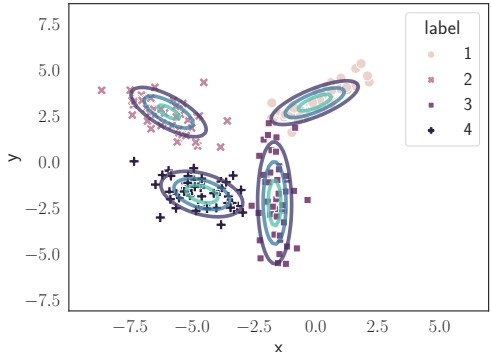

Figure 5: Example of a dataset sampled with the procedure outlined in Appendix E.1. The only difference from the dataset in the experiment is the dimensionality and number of clusters which was minimized for visualization purposes.

| Setting | Value |
|---|---|
| Runs | 5 |
| Iterations | 500 |
| Batch Size | 32 |
| Optimizer | SGD+Momentum(0.97) |
| Learning Rate | 0.001 |
| Teacher Momentum ($\alpha$) | 0.9 |
| Masking Ratio | 0.5 |

Table 8: **Linear Model Experiment Settings**

### E.2 IMPLEMENTATION DETAILS

We implement RC-MAE based on our baseline MAE (He et al., 2022) by adding EMA Teacher which has the same architecture as student (*i.e.*, MAE). To be specific, the EMA teacher consists of an encoder (*e.g.*, ViT-B) and a decoder (Transformer blocks), following the same number of blocks and dimension size as MAE. RC-MAE with ViT-S has 12 blocks of Transformer blocks with a dimension size of 384 for an encoder and 4 blocks of Transformer blocks with a dimension size of 256 for a decoder. RC-MAE with ViT-B has 12 blocks of Transformer blocks with a dimension size of 768 for an encoder and 8 blocks of Transformer blocks with a dimension size of 512 for a decoder. RC-MAE with ViT-L has 24 blocks of Transformer blocks with a dimension size of 1024 for an encoder and 8 blocks of Transformer blocks with a dimension size of 512 for a decoder. For pre-training, we optimize only the student network including an encoder and a decoder by the gradients while updating the teacher network (encoder & decoder) by EMA of the student weights as shown in Fig. 1(b). For downstream tasks, we only use the pre-trained encoder part from the teacher network as other self-distillation methods ((Caron et al., 2021; Assran et al., 2022; Zhou et al., 2022; Dong et al., 2022)). We also test which one is better between student and teacher for transfer learning in Appendix F.1.

### E.3 IMAGENET EXPERIMENTS

We follow the implementation details of the official MAE (He et al., 2022) code[1] for all pre-training, fine-tuning and linear probing. While He et al. (2022) used 128 TPU-v3 cores, we have tried to reproduce the baseline MAE and train our RC-MAE on the same local GPU environment, which has 8 NVIDIA V100 GPUs (32GB) for more accessibility in the community. Although the authors provide the guideline using 8 nodes with 8-GPUs (64-GPUs) in their official code written in `Pytorch`, we train RC-MAE and MAE on the same 8-GPUs environment. To do this, we use `accum_iter`

---

[1]https://github.com/facebookresearch/mae

Table 9: **Pre-training setting.**

| Config | Value |
|---|---|
| optimizer | AdamW (Loshchilov & Hutter, 2019) |
| optimizer momentum | $\beta_1, \beta_2 = 0.9, 0.95$ (Chen et al., 2020a) |
| weight decay | 0.05 |
| base learning rate | 1.5e-4 |
| batch size | 4096 |
| learning rate schedule | cosine decay (Loshchilov & Hutter, 2016) |
| warmup epochs | 40 |
| augmentation | RandomResizedCrop |
| mask ratio | 75% |

Table 10: **End-to-end fine-tuning setting.**

| Config | Value |
|---|---|
| optimizer | AdamW |
| optimizer momentum | $\beta_1, \beta_2 = 0.9, 0.999$ |
| weight decay | 0.05 |
| layer-wise lr decay | 0.75 (S), 0.65 (B,L) |
| base learning rate | 5e-4 |
| drop path | 0.1 (S,B), 0.2 (L) |
| batch size | 1024 |
| learning rate schedule | cosine decay |
| warmup epochs | 5 |
| training epochs | 300 (S), 100 (B,L) |
| augmentation | RandAug(9, 0.5) |
| label smoothing | 0.1 |
| mixup | 0.8 |
| cutmix | 1.0 |

Table 11: **Linear probing setting.**

| Config | Value |
|---|---|
| optimizer | LARS |
| base learning rate | 0.1 |
| weight decay | 0 |
| batch size | 16,384 |
| learning rate schedule | cosine decay |
| warmup epochs | 10 |
| training epochs | 90 |
| augmentation | RandomResizedCrop |

which the authors provide for accumulating gradients. However, when using `accum_iter`, a re-producing issue[2][3] occurs in that the re-produced result (MAE-ViT-B) achieves slightly lower performance *e.g.*, for ViT-B, 84.4% vs. 84.6% in end-to-end fine-tuning as shown in Table 4.

**Pre-training.** Thanks to `accum_iter`, we can set the same effective batch size of 4096 as MAE, *e.g.*, 256 (batch_size_per gpu) × 8 (GPUs) × 2 (accum_iter) for both ViT-B and ViT-S. For ViT-S, we set 512 (batch_size_per gpu) × 8 (GPUs) × 1 (accum_iter). we follow the linear learning rate scaling rule (Goyal et al., 2017): $lr = base\_lr \times batch\_size/256$. The detail settings are shown in Table 9.

**End-to-End fine-tuning.** Since MAE (He et al., 2022) does not have a ViT-S result, we train MAE and RC-MAE using ViT-S following the training protocol in BEiT (Bao et al., 2022). For ViT-S, we use a layer-wise lr decay of 0.75, a stochastic drop path of 0.1, and a fine-tuning epoch of 300. We can expect that a more suitable hyper-parameter search boosts performance. For ViT-L, we can set the effective batch size to 1024 by using `accum_iter`, *e.g.*, 64 (batch size per gpu) × 8 (GPUs) × 2 (accum_iter). The detail settings are shown in Table 10.

**Linear probing.** We use LARS (You et al., 2017) optimizer. The detail settings are shown in Table 11.

**Robustness Evaluation on ImageNet-variants** We use the same model weights (RC-MAE w/ViT-B for 1600epoch) fine-tuned on the original ImageNet-1K as shown in Table 4 and only test without any specialized fine-tuning on the different validation sets, such as ImageNet-C (Hendrycks & Di-

---

[2]https://github.com/facebookresearch/mae/issues/30
[3]https://github.com/facebookresearch/mae/issues/91

Table 12: **Comparison between Student vs. Teacher networks** on ImageNet-1K evaluation. FT and LN denote fine-tuning and linear probing top-1 accuracies, respectively.

| Checkpoint | FT | | LN | |
|---|---|---|---|---|
| | top-1 | top-5 | top-1 | top-5 |
| Student | 83.48 | 96.59 | 68.38 | 87.88 |
| Teacher | **83.58** | **96.64** | **68.41** | **87.94** |

Table 13: **Total GPU memory usage and throughput** using ViT-L backbone. `OOM` represents that it was not able to run on 8 V100-32GB GPUs due to "Out of memory".

| | GPU Memory Usage (GB) | | | | Throughput (imgs/sec.) | | | |
|---|---|---|---|---|---|---|---|---|
| Batch size | 128 | 256 | 512 | 1024 | 128 | 256 | 512 | 1024 |
| MAE (He et al., 2022) | 84 | 103 | 140 | 213 | 533 | 853 | 1191 | 1442 |
| MSN (Assran et al., 2022) | 183 | OOM | OOM | OOM | 78 | OOM | OOM | OOM |
| iBOT (Zhou et al., 2022) | 227 | OOM | OOM | OOM | 123 | OOM | OOM | OOM |
| BootMAE (Dong et al., 2022) | 98 | 116 | 154 | 225 | 376 | 557 | 701 | 813 |
| **RC-MAE** | **95** | **114** | **152** | **226** | **441** | **674** | **914** | **1101** |

etterich, 2019),-A (Hendrycks et al., 2021b),-R (Hendrycks et al., 2021a), and -Sketch (Wang et al., 2019).

### E.4 OBJECT DETECTION AND SEGMENTATION ON COCO

Following the training protocol (Li et al., 2021) and implementation details as MAE (He et al., 2022), we fine-tune Mask R-CNN (He et al., 2017) with a ViT-Base backbone COCO dataset. To adapt the vanilla ViT with the isotropic structure (*i.e.*, single-scale) for FPN (Lin et al., 2017) in the backbone of Mask R-CNN, we equally divide the ViT into four blocks and apply convolutions to downsample or upsample the intermediate features for building the multi-scale feature pyramid. For a fair comparison with MAE, we train both MAE and the proposed RC-MAE for 100 epochs with the same training protocol on the same 8 GPU environment: large-scale jitter (LSJ), AdamW (Loshchilov & Hutter, 2019) with half-periodic cosine learning rate decay, linear warmup, and drop path regularization (Huang et al., 2016) (*e.g.*, 0.1 for ViT-Base). We use a batch size of 16 (2 per GPU), a learning rate of 4e-5, and a weight decay of 0.1. We implement models based on the reproduced `Pytorch` code[4,5].

## F ADDITIONAL EXPERIMENTS

### F.1 STUDENT VS. TEACHER NETWORKS.

To determine which is better between student and teacher networks for transfer learning, we conduct fine-tuning and linear probing on ImageNet-1K by using each pre-trained weight. Table 12 shows the comparison between student and teacher networks. We find that the results of the teacher network are slightly higher than those of the student, which is similar to DINO (Caron et al., 2021). Thus, if unspecified, we use teacher network weight for downstream tasks.

### F.2 RESOURCE COMPARISON: MEMORY & COMPUTATION COST

In terms of memory consumption and computation cost, we conduct resource comparison with the baseline method MAE [6] (He et al., 2022) and the self-distillation-based MIM methods, *e.g.*,

---

[4] https://github.com/hustvl/MIMDet

[5] At the time of submission of this paper, the official code of benchmarking ViT detection (Li et al., 2021) was not released.

[6] https://github.com/facebookresearch/mae

MSN [7] (Assran et al., 2022), iBOT [8] (Zhou et al., 2022), and BootMAE[9] (Dong et al., 2022). We compare GPU memory usage and throughput while pre-training with ViT-L/16 on 8 V100-32GB GPUs as shown in Table 13. Memory usage is directly measured by using `nvidia-smi` command. Throughput is an average speed (images/second) for an epoch. We follow the original configuration for each method except batch size for a fair comparison. Batch sizes from 128 to 1,024 were employed to observe the scalability of the methods. We note that iBOT (Zhou et al., 2022) and MSN (Assran et al., 2022) could not run with batch sizes from 256, while other methods could deal even with a batch size of 1,024. iBOT feeds both masked and unmasked patches into the student encoder by following the masked image modeling employed in BeiT (Bao et al., 2022). Therefore, it requires $4\times$ memory usage of MAE-based methods for the student encoder. In addition, for further performance gain, iBOT exploits multiple local crops. Although we removed the local crops, memory usage was still 164GB which is much higher than MSE-based methods. MSN takes a lower masking ratio than other methods (0.5 vs. 0.75), resulting in $2\times$ memory consumption for the student encoder.

---

[7] https://github.com/facebookresearch/msn
[8] https://github.com/bytedance/ibot
[9] https://github.com/LightDXY/BootMAE

