# OpenReview forum: "Exploring The Role of Mean Teachers in Self-supervised Masked Auto-Encoders"
_ICLR.cc/2023/Conference — ICLR 2023 poster_

### Official Review · Reviewer_5RHA · 2022-10-19

**Confidence:** 4
**Correctness:** 2
**Technical Novelty And Significance:** 2
**Empirical Novelty And Significance:** 2
**Recommendation:** 5

**Clarity, Quality, Novelty And Reproducibility:**

**Clarity.**  The paper proposes a simple extension of MAE, that gets complexified by the evaluation of the gradients.  The explanation of the expansion of the gradients and the memory bank are hard to follow from the main description.  After several reads of the appendix one can figure out the details though.  The readability could be improved if more explanations were provided in the main text.

**Reproducibility.** The paper is heavily inspired by MAE (He et al., 2022).  The details of the implementation are left open, and it is described that He et al.'s implementation was followed.  It will be good for the sake of reproducibility to detail the paper setup, regardless of point the reader to He et al.  Moreover, it is not clear if the pre-training includes the decoder and the reconstruction of the image on top of the patches reconstruction.

**Novelty.** The inclusion of a consistency and reconstruction loss over an asymmetric siamese network is not new.  The application to patches may be new.  The idea of masking and denoising as a source of supervision is not new as well.  The benefits seem to come from the ViT that helps to take advantage of the patches structure.  However, that was proposed before by He et al.

The expansion of the gradient and trying to understand the bank of gradients is interesting, and I'm not aware of others doing a similar exploration.  However, the link between that exploration and the proposed loss avoids me.

**Strength And Weaknesses:**

Strengths:
- The paper uses a simple extension to the mask autoencoders with ViT (MAE) by adding a consistency loss over an asymmetric siamese setup (Teacher-Student), following the success of recent asymmetric self-supervised models that rely on exponential moving average for the update of the other stream.
- The results show improvement over the existing MAE, and two approaches for masking are explored and evaluated.
- The role of the teacher as a memory bank of past gradients is explored.  A simple experiment is performed on a linear version of the Teacher-Student setup that shows the changes on the gradient w.r.t. the similarity of the features.  The same experiment is repeated with less explainability with more complex models.

Weaknesses:
- The claims of using the "role of the teacher" as basis to propose the extension of the model are not clear.

  In particular, the proposed loss of reconstruction and consistency is straightforward on the multi-stream setup.  In its current form, the paper reads as adding more complexity by trying to explain the bank of gradients as a basis for the extra consistency term in (6).  While the result is interesting, it doesn't seem to resonate with the proposal of updating the MAE.

  Consider a simpler and cleaner approach of introducing the consistency and reconstruction loss, instead of adding unnecessary complexity to justify the claims.

- It is not clear from the explanation in the Appendix how well the projection ($S\hat{x}$) and the difference of the agreement between the teacher and student ($(S-T)\hat{x}$) relate to the original data ($x$) also influence the final gradient.

  Wouldn't errors in the projection or difference between the streams also influence the gradients?  And that will also make a difference on the bank and what is stored in it?

  Should the generic linear model also contain a term that shifts the vector $\Theta \tilde{x} -x$ depending on the difference of the streams as well?

**Summary Of The Paper:**

The paper extends the Masked Auto-Encoders (MAE) with Vision Transformers by incorporating a Teacher-Student setup (asymmetric siamese model with exponential moving average update for one of the streams).  The proposal uses a reconstruction loss over the masked patches and adds a consistency loss over the reconstruction of both streams.  The experiments show an improvement over MAE in ImageNet-1K classification, and COCO detection and segmentation downstream tasks.

The paper also frames the Teacher-Student setup as a memory of past features that are used by the student to follow the gradient of the previous samples' features.  The analysis was done over a linear simplification of the Teacher-Student model, and an experiment shows the gradient responses on different cases of similarity and dissimilarity of the samples.

**Summary Of The Review:**

The paper proposes a simple yet effective loss based on the reconstruction of patches using ViT as an encoder, and a two-stream asymmetric model with exponential moving average as update for one of the streams (teacher-student) that should agree in their reconstructions (consistency loss).  The claim of the authors of using the understanding of the teacher as a bank of gradients to propose the above mentioned setup is beyond me.

The setup is complicated, and tries to tie to topics and package them together.  Perhaps a simpler write-up with these two pieces presented individually could be easier to follow and allow them to shine.

The details about the implementation (encoders, decoders, and setup) is not presented.  The paper references the same implementation as He et al.'s setup, yet the differences for this proposal are not explained.

---

> ### Author Response · Authors · 2022-11-08
> **Author Response (2/2)**
>
> >*The inclusion of a consistency and reconstruction loss over an asymmetric siamese network is not new.*
>
> - We would like to note that the novelty in our work lies in the analysis of the student/teacher dynamics and not in the actual siamese network setup itself.
> - We would also like to clarify that our model is **symmetric** and not asymmetric. This misunderstanding may stem from our related work section, where we state that the theoretical analysis of self-supervised pre-training with asymmetric models such as BYOL has been an interesting recent development. This inspired us to look for more basic explanations into the roles of EMA teachers, where we found surprisingly _little_ analysis even though EMA teachers are common in SSL.
> - We state in section 5 that in RC-MAE the teacher is the **same** architecture as the student.
>
> > *I'm not aware of others doing a similar exploration. However, the link between that exploration and the proposed loss avoids me.*
>
> - In our exploration, we were able to draw conclusions about the extra information supplied in the consistency gradient coming from the teacher. We then tried to observe these effects in a linear model (Fig. 2) and further observe these effects in the full RC-MAE (a ViT) in Fig. 3. Therefore, we used the findings from our exploration to observe these effects in a simple, effective, and modern SSL method (RC-MAE), which provides the link between the exploration and the RC-MAE results in our work.
>
> Thank you for reading this response. We would be happy to discuss any further questions or concerns!

---

> ### Author Response · Authors · 2022-11-08
> **Author Response (1/2)**
>
> Thank you for your valuable comments and appreciation for our work. Below we discuss the points you have raised in detail.
>
> > *the paper reads as adding more complexity by trying to explain the bank of gradients as a basis for the extra consistency term in (6). While the result is interesting, it doesn't seem to resonate with the proposal of updating the MAE. Consider a simpler and cleaner approach of introducing the consistency and reconstruction loss, instead of adding unnecessary complexity to justify the claims.*
>
> - Thank you for noting that the result is interesting. We would like to point out that the main purpose of our work is in the analysis of the EMA teacher in modern SSL, and we are not adding more complexity by trying to justify the addition of a teacher to the MAE.
>   - For clarification, could you let us know specifically which part you feel is complex?
> - As the title of our paper suggests, we aimed to explore the role of EMA teachers, and RC-MAE is a simple instantiation of such a student/teacher model which we use to analyze and observe our theoretical findings.
>
> >*It is not clear from the explanation in the Appendix how well the projection ($S\tilde{\mathbf{x}}$ ) and the difference of the agreement between the teacher and student (($S-T\tilde{\mathbf{x}}$ ) relate to the original data ($\mathbf{x}$) also influence the final gradient. Wouldn't errors in the projection or difference between the streams also influence the gradients? And that will also make a difference on the bank and what is stored in it?*
>
> - Yes, any errors in the projection $S\tilde{x}$ (the original reconstruction objective) will definitely have an impact on what is stored in the gradient memory (the teacher) because the teacher is composed of previous students.
> - Our analysis uncovers how the teacher uses this information when providing the consistency objective to the student. The alignment with the original data is outside the scope of our analysis, as we assume the reconstruction objective provides a general trajectory towards a solution to the original data.
> - We aimed to uncover how the additional consistency objective affects the training procedure. For example, the work done in the original MAE provided evidence that an MAE provides an appropriate solution, so in our work, we analyzed the dynamics of the *_augmentation_* that the teacher provides to this established training procedure.
>
> >*Should the generic linear model also contain a term that shifts the vector $\Theta\tilde{x}-x$ depending on the difference of the streams as well?*
>
> - We are not sure what difference in the streams is being referred to. We defined $\tilde{\mathbf{x}}$ to be the masked input. There is therefore no difference in the inputs (streams) to the student and the teacher.
>
> >***Clarity.** The paper proposes a simple extension of MAE, that gets complexified by the evaluation of the gradients.*
>
> - We would like to stress that we are NOT evaluating the gradients in such a way to justify or add complexity to our model. Our aim is to explain the mechanism by which EMA teachers aid in the training process.
> - To our knowledge, this has not been adequately explained in previous literature, and it is applicable beyond RC-MAE. In RC-MAE, we add an EMA teacher to the MAE model as a simple extension in order to evaluate the dynamics and attempt to observe the predictions from the linear model.
> - **Fig. 3** shows that this is indeed the case, and **the predicted effects can be observed in the full RC-MAE model (a ViT)**.
>
> >***Reproducibility.** The paper is heavily inspired by MAE (He et al., 2022). The details of the implementation are left open, and it is described that He et al.'s implementation was followed.*
> - Due to space constraints, we described the training/evaluation setup details in Appendix E. As you mention, we reference He et al. because we use MAE as a baseline and use an identical network for the EMA teacher.
> - For pre-training, we optimize only the student network including an encoder (e.g., ViT-B) and a decoder (e.g., Transformer blocks) by the gradients while updating the teacher network (encoder+decoder) by EMA of the student weights as shown in Fig.1(c). For downstream tasks, we only use the pre-trained encoder part from the teacher network as other self-distillation methods (DINO, MSN, iBOT, and BootMAE). We have added these details to Appendix E.2 in the revised version for completeness.
> - We would like to note that we think we have covered all relevant training details. Additionally, all code was included in the supplementary files. We would be happy to add any specific further details to Appendix if the reviewer feels anything is missing.

---

> > ### Comment · Reviewer_5RHA · 2022-11-09
> > **Reply after reading the comments**
> >
> > **Regarding complexity**
> >
> > My comment refers to the justification for the proposal of the RC-MAE (6).  It reads as a simple yet direct approach that could be evaluated and have interesting results.  However, the justification over the gradient analysis and then generalizing to the non-linear case seems to much of a stretch.
> >
> > In your rebuttal, the contribution that you want to put forward is the analysis and not the model, though.  I think this weakens your paper, and I would still argue that the model is a contribution.  I just don't think that the way of reaching it (as it is proposed in the current paper) is the best.
> >
> > **Gradient analysis**
> >
> > Thanks for providing extra explanation about the gradient.
> >
> > When I referred to the "difference in the streams" I was talking about the difference $S-T$ between the student and teacher in the equation after (8) in Appendix B.1.1.
> >
> > It is still not clear how this difference can be neglected since the gradient is interpreted to be moving in the direction of $\Theta \tilde{x} - x$ but there is still a term that depends on the difference $S-T$.  How is that neglectable?  Am I missing something in your simplification?
> >
> > (*As a side note, this is precisely why it is a good practice to number every equation even if not referred to in the original text.  You may not use it, but further readers may need to refer to your work.  It is much simpler to use the original numbers than the references of the existing ones, as I did above.*)
> >
> > **Overall evaluation**
> >
> > Despite the efforts explaining the method, I still find the paper contribution limited.  More now that the rebuttal claims that the main contribution is the linear simplification of the models to produce a bank of gradients as the explanation of the function of the models.  However, the existing methods are non-linear and no explanation or analysis is given in this regard.  I understand that the complexity is a limitation, but that is the problem at hand.
> >
> > Perhaps, this is a first step towards that direction.  But, I think that the paper needs more work in that regard for this to be the main contribution.
> >
> > If, on the other hand, the proposed model RC-MAE is considered the contribution, then the work in re-writing the paper and explaining the proposal in a more straightforward way will be the main drawback and limitation.
> >
> > Either way, I still find the manuscript not ready for publication.

---

> > > ### Author Response · Authors · 2022-11-09
> > > **Clarifications**
> > >
> > > Thank you for your continued discussion. We feel there is a misunderstanding, which we will attempt to address below.
> > >
> > > > *However, the justification over the gradient analysis and then generalizing to the non-linear case seems to much of a stretch.*
> > >
> > > >*The existing methods are non-linear and no explanation or analysis is given in this regard. I understand that the complexity is a limitation, but that is the problem at hand.*
> > >
> > > - Please note that we **have observed** the dynamics in the **full RC-MAE (a ViT)** in Fig. 3, which is a large non-linear model.
> > > - The magnitude and direction of the gradients in Fig. 3 are conditional based on the feature similarity, as predicted by our analysis and observed in the linear model.
> > > - This is indicated in section 4, the caption of Fig. 3, and in our response to your initial comments under the **clarity** point.
> > >
> > > ---
> > >
> > > >*It is still not clear how this difference can be neglected since the gradient is interpreted to be moving in the direction of $\Theta\tilde{x} - x$, but there is still a term that depends on the difference $S - T$. How is that neglectable? Am I missing something in your simplification?*
> > >
> > > - We think this is a misunderstanding regarding the text at the bottom of page 15.
> > > - The part of the gradient which contains the $S - T$ term (which is $(S - T)\tilde{x}\tilde{x}^\top$) **is not neglected**, and is in fact what the consistency loss contributes to the gradients. **This is precisely what we analyze throughout our work.**
> > > - The generic gradient which was expressed as $\Theta\tilde{x} - x$ at the bottom of page 15 of the appendix was meant to signify a generic 'matching' loss gradient (i.e. either the reconstruction or the consistency loss gradient)
> > > - For example, the reconstruction loss gradient of the linear model is $(S\tilde{x} - x)\tilde{x}^\top$ and the consistency loss becomes $(S\tilde{x} - T\tilde{x})\tilde{x}^\top$ where $T\tilde{x}$ is a stand-in for $x$ in the consistency loss.
> > > - Therefore, the conclusion we come to at the end of page 15 is that the correction provided to $S$ *by the consistency loss* (i.e. the $(S - T)$ term) is conditionally weighted based on the dot product $\hat{x}^\top\tilde{x}$ and the final projection of the current feature $\tilde{x}^\top$
> > > - This can be seen in the two variables which are on the RHS of equation 12 (numbered in the updated text). **Note that the RHS of equation 12 is derived directly from the $(S - T)$ term in the linear model, which is what we are analyzing.**
> > > - We have updated the text near the bottom of page 15 of the appendix to clarify this point. Please let us know if there is anything we can do to make it clearer.
> > >
> > > ---
> > >
> > > >*the contribution that you want to put forward is the analysis and not the model.*
> > >
> > > - The main purpose of our work as the title suggests, is exploring the role of EMA teachers which are commonly used without a deep understanding of what their contributions are. Therefore, the model is part of the contribution in that it demonstrates *what* gains can be realized by adding a teacher and our analysis describes *how* they are realized. We feel that the *how* question has been particularly neglected in previous works, which is why we chose to direct our attention there.
> > >
> > > ---
> > >
> > > >*it is a good practice to number every equation*
> > >
> > > - We have updated all equations to be numbered, as suggested.
> > >
> > > ---
> > >
> > > Thank you for your continued review, we remain open to further discussion.

---

> > > > ### Comment · Reviewer_5RHA · 2022-11-17
> > > > **Still not convinced, and confused of the overall description**
> > > >
> > > > I thank the authors for the effort and time responding to my comments.
> > > >
> > > > I still have problems following your narrative, and I'm not convinced of the main contributions of the paper.  And, **I still find it not ready for publication**.
> > > >
> > > > **Details**
> > > >
> > > > On one hand, there is the simple RC-MAE proposal that was diminished as a side contribution.  Yet, I see this as one interesting feature.  The proposal is simple: explore the pixel space for the reconstruction and consistency losses that were explored before by others in SSL (e.g., BYOL, DINO, MSN).  The main problem, as I stated before, is that this presentation is convoluted in the current manuscript.  But you are the ones that decide how do you want to present your findings.
> > > >
> > > > On the other hand, there is a simplification to understand the EMA of the teachers on a linear case.  While the results are interesting, I do not see them generalizing.  Despite your efforts of Fig. 3, I do not see the same trends you mentioned.
> > > >
> > > > For instance, while the linear case, Fig. 2(a), clearly shows that the consistency loss diminishes as iterations move forward for the three cases of consecutive inputs (same, similar, and different), the non-linear case, Fig. 3(a), shows that they grow and then suddenly drop.  The problem is that the results do not explain the non-linear case, and as such several questions arise from this lack of evidence: How is this behavior explained by the memory alone?  Is this sudden drop due to some luck in the exploration phase?  Would this behavior happen again on a different batch of samples?  I don't see these questions been addressed by the discussion you present.  In this regard, there is need of more thorough empirical evaluation to understand the non-linear case.  And, most probably, you will need to perform several runs and show average (with std. deviations) of the results.
> > > >
> > > > Similarly, the reconstruction loss of the linear case, Fig. 2(b), clearly shows how the different samples respond differently from the student and teacher.  Yet, in the non-linear case, Fig. 3(b), all the cases present the same trend.  My guess is that the student and the teacher are having the same reconstructions and not learning the patches in a different way.  And again, the same questions and problems as above arise.  In particular, how can you explain this with the memory bank?
> > > >
> > > > **Continue working on it**
> > > >
> > > > Once again, *I think the work is interesting and worth pursuing*.  I encourage you to continue with the non-linear exploration.  The simplification that you show is an interesting first step, but more work is required to make it publishable.  As I mentioned in my original evaluation, I found the descriptions and the narrative hard to follow.  The eagerness of present a result (RC-MAE) and compare it against the literature may be interfering with your descriptions.  Perhaps deciding on one way or the other will help to streamline your descriptions and explanations.  Then, you also need to design a set of robust empirical experiments to validate your approximation on the non-linear case in a convincing way, and more importantly, in a thorough one.
> > > >
> > > > **Minor comments**
> > > > - There seems to be a discrepancy between the cases in Table 1 and the ones presented in Proposition 4.1 (and the appendix).  Case 2, in the proposition, refers to the orthogonal case.  Thus, it should be for orthogonal or different elements.  Yet, in Table 1, the different inputs is Case 3.
> > > > - Define $\Theta$ in the discussion in the proof B.1.1.  The change of variable convolutes the argument.
> > > > - I would number all the individual equations, even the ones within the group of equations from the proofs.  That way it is easier for others to reference them.  In case you don't want to number them individually, you could use a sub-equation number or identifier.

---

> > > > > ### Author Response · Authors · 2022-11-18
> > > > > **Clarifications And Further Discussion**
> > > > >
> > > > > Thank you for the continued discussion:
> > > > >
> > > > > ---
> > > > >
> > > > > > *For instance, while the linear case, Fig. 2(a), clearly shows that the consistency loss diminishes as iterations move forward for the three cases of consecutive inputs (same, similar, and different),*
> > > > >
> > > > > - We must first clarify that **Fig. 2(a) does not show the loss value**, but the norm of the gradients of the loss $\Vert \nabla \mathcal{L}\Vert$.
> > > > > - Fig 2(a) has a decreasing gradient norm because the linear model converges rather quickly and therefore the norm of the gradient goes down. However the order of the lines is preserved, which is what our analysis predicts.
> > > > >
> > > > > ---
> > > > >
> > > > > > *The non-linear case, Fig. 3(a), shows that they grow and then suddenly drop. The problem is that the results do not explain the non-linear case, and as such several questions arise from this lack of evidence:*
> > > > >
> > > > > >*How is this behavior explained by the memory alone? Is this sudden drop due to some luck in the exploration phase?*
> > > > >
> > > > > - **The drop in the gradient norm** that is present at the end of Fig 3(a) is due to the decreasing learning rate in the default MAE model towards the late epochs which causes the teacher and student to be closer together and therefore have a smaller gradient.
> > > > > - **The behavior which is explained by the 'memory bank' interpretation is:**
> > > > >   1. **The order of the lines concerning the norm of the gradient of the consistency loss. (Figs.2-3(a))**
> > > > >   2. **The directions of the gradients being commensurately opposite (Figs. 2-3(c))**
> > > > > - This above two points are predicted by our analysis, and consistently observed throughout training, on random samples of data, in a wide variety of changing learning rates, levels of convergence, and model types (linear and ViT).
> > > > >
> > > > > ---
> > > > >
> > > > > >*Would this behavior happen again on a different batch of samples?*
> > > > >
> > > > > - **Yes, it does happen for different batch of samples.** In Fig. 3, the results were computed on a batch of 512 total samples.
> > > > > - For each checkpoint, a random batch of 512 samples from the dataset is selected, which means that each point on the line plot is a different random sample of 512 instances.
> > > > > - If there was high variance between batches, the lines would be noisy and 'crisscrossing,' which is not the case. Instead we witness a steady, predictable order of norms.
> > > > >
> > > > > ---
> > > > >
> > > > > > *Similarly, the reconstruction loss of the linear case, Fig. 2(b), clearly shows how the different samples respond differently from the student and teacher. Yet, in the non-linear case, Fig. 3(b), all the cases present the same trend. [...] And again, the same questions and problems as above arise. In particular, how can you explain this with the memory bank?*
> > > > >
> > > > > - **Figs. 2-3(b) show the reconstruction loss** so they do show any direct effect of the memory bank.
> > > > > - The point of Fig. 2-3(b) is to show the difference between the reconstruction and consistency loss. **Note that we make no prediction or claim about the norm of the gradient of the reconstruction loss, and Figs. 2-3(b) are presented for completeness.**
> > > > >
> > > > > ---
> > > > >
> > > > > > *Fig. 2-3(b): my guess is that the student and the teacher are having the same reconstructions and not learning the patches in a different way.*
> > > > >
> > > > > - **If this was true, then there would be no benefit to using a teacher.**
> > > > > - Many SSL works use teachers of this form, and realize a benefit, including our RC-MAE.
> > > > > - Previous works also *guessed* about the reasons why teachers help, with no mechanism or analysis provided.
> > > > > - This is why we are studying: *How the teacher helps during training,* which has not been adequately described before
> > > > >
> > > > > ---
> > > > >
> > > > > > *The proposal is simple: explore the pixel space for the reconstruction and consistency losses that were explored before by others in SSL (e.g., BYOL,DINO,MSN).*
> > > > >
> > > > > - BYOL, DINO, and MSN are based on feature space, not pixel space. Specifically, they use global representation (e.g., embedding vector), while MAE and RC-MAE use low-level representations (pixel-level). Moreover, they don't have the reconstruction task, only a consistency loss. We analyze the effectiveness of EMA teacher on the reconstruction task of masked auto-encoder architecture, which has not been well explored yet except for BootMAE which we addressed in the Response to Reviewer `rLZG` and the Shared response.
> > > > >
> > > > > ---
> > > > > ## Updates in the Revised Text
> > > > >
> > > > > > The cases in Table 1 are out of order compared to Section 4 and the Appendix
> > > > >
> > > > > Thank you. We have corrected the order in Table 1.
> > > > >
> > > > > > Define Θ:  in the discussion in the proof B.1.1. The change of variable convolutes the argument.
> > > > >
> > > > > We have updated $\Theta$ to refer to a generic parameter of the model.
> > > > >
> > > > > > I would number all the individual equations, even the ones within the group of equations from the proofs.
> > > > >
> > > > > All lines of all equations in the appendix have been numbered.
> > > > >
> > > > > ---
> > > > >
> > > > > **Thank you for your suggestions, we have incorporated them in the revision to improve our work. We hope we have cleared up some concerns, and we remain open to discussion until the very end of the discussion period.**

---

> > > > > > ### Comment · Reviewer_5RHA · 2022-11-28
> > > > > > **Thanks for the additional information, but I'm still not convinced**
> > > > > >
> > > > > > I thank the authors for the effort explaining their method once more.
> > > > > >
> > > > > > I still do not understand the point of the gradient of the losses.  Now, the authors mentioned that is the order of the losses that matter, and not the relation of the convergence with the memory bank.  However, I do not see how there is an order relation between the gradients magnitude and the three cases from their proof of proposition 4.1.  Where does the order relation comes from?
> > > > > >
> > > > > > I disagree that the reconstruction has no impact on the teacher and student.  The idea of the reconstruction loss is to use the visual similarity of the encoded features to improve the embeddings.  If the decoder is not passing the reconstruction loss back to the models then it is not helping in any way.  I guess that from Fig. 1(c), the gradients from the reconstruction loss are backpropagated to the student, which in turns passes them to the teacher through the EMA.  In that sense, the changes of the reconstruction loss (Figs. 2-3(b)) show these behavior.  Note that my guess of having the decoder for the teacher to produce similar outputs from the decoder of the student is related to the EMA on it as well.  In this regard, shouldn't the same arguments of a memory bank also be applied to the decoder as well?  And so, then one should expect to see the similar order in Figs. 2-3(b)?  (Note again that I do not understand where the order you mention comes from.)
> > > > > >
> > > > > > Once more, I restate that I find the manuscript interesting, but not ready for publication.  It has several flaws that need to be addressed and ironed out.

---

> > > > > > > ### Author Response · Authors · 2022-11-28
> > > > > > > **Clarifying Misunderstandings**
> > > > > > >
> > > > > > > ### Thank you for continuing the discussion. We think there are still misunderstandings about our work which we will attempt to address below:
> > > > > > >
> > > > > > > ---
> > > > > > >
> > > > > > > > *I still do not understand the point of the gradient of the losses. Now, the authors mentioned that is the order of the losses that matter.*
> > > > > > >
> > > > > > > - We must point out again that Figs. 2-3 are analyzing the norm of the loss gradient and not the loss. Therefore, it is not the order of the losses, but the order of the gradient norms that matters.
> > > > > > > - In the original version, we stated the following in the '**Expected Outcome**' paragraph on p. 5 :
> > > > > > >   - *"we would expect the consistency loss to produce a larger gradient for the $\textcolor{blue}{\textbf{same}}$ or $\textcolor{orange}{\textbf{similar}}$ inputs and a smaller gradient for $\textcolor{green}{\textbf{different}}$ inputs.*"
> > > > > > > - Additionally, the captions of Figs 2-3 in the original submission stated the following when referring to subfigure a:
> > > > > > >   - *"$\Vert \nabla_S \mathcal{L}_c \Vert$ is larger when $\text{cos}(\hat{x}, \tilde{x})$ is larger due to Eq. 5"*
> > > > > > >
> > > > > > > ---
> > > > > > >
> > > > > > > > *However, I do not see how there is an order relation between the gradients magnitude and the three cases from their proof of proposition 4.1. Where does the order relation comes from?*
> > > > > > >
> > > > > > > - As stated in the captions of Figs. 2-3(a), the order relation comes from Eq. 5.
> > > > > > > - The RHS of Eq. 5 gives an upper bound for the norm of the gradient.
> > > > > > > - The models for all three cases share the same gradient history up until the point where we run the test in Figs. 2-3, and so, the first term in the sum is the only term which will differ between the three cases in Figs. 2-3.
> > > > > > > - Therefore, the upper bound on the gradient norm given by Eq. 5 should vary according to the dot product with the previous inputs $\mathbf{\hat{x}}^\top \mathbf{\tilde{x}}$ (a lower magnitude lowers the upper bound and higher magnitude raises the upper bound.)
> > > > > > > - Figs. 2-3 do show this in both cases. $\textcolor{blue}{\textbf{blue}} > \textcolor{orange}{\textbf{orange}} > \textcolor{green}{\textbf{green}}$ on different random samples, on both linear and ViT models, and consistently throughout the training checkpoints.
> > > > > > > - Note that **Same** (Case 1:$\textcolor{blue}{\textbf{blue}}$, high magnitude dot product), **Similar** (Case 3: $\textcolor{orange}{\textbf{orange}}$, medium magnitude dot product), **Different** (Case 2: $\textcolor{green}{\textbf{green}}$, low magnitude dot product)
> > > > > > > - It is also important to note that the latter epochs of Fig. 2(a) follow the same order, and we can include a close-up of that part in the final version.
> > > > > > >
> > > > > > > ---
> > > > > > >
> > > > > > > > *I disagree that the reconstruction has no impact on the teacher and student. The idea of the reconstruction loss is to use the visual similarity of the encoded features to improve the embeddings. If the decoder is not passing the reconstruction loss back to the models then it is not helping in any way. I guess that from Fig. 1(c), the gradients from the reconstruction loss are backpropagated to the student, which in turns passes them to the teacher through the EMA. In that sense, the changes of the reconstruction loss (Figs. 2-3(b)) show these behavior.*
> > > > > > >
> > > > > > > - We are not sure why there is a disagreement on this point. We never state nor claim that the reconstruction loss has no impact on the student and teacher. The reconstruction loss gives the main gradient signal. We analyze the contributions of the additional gradient signal ($\nabla_S \mathcal{L}_c$) which is provided by the teacher.
> > > > > > >
> > > > > > > ---
> > > > > > >
> > > > > > > > *Note that my guess of having the decoder for the teacher to produce similar outputs from the decoder of the student is related to the EMA on it as well. In this regard, shouldn't the same arguments of a memory bank also be applied to the decoder as well?*
> > > > > > >
> > > > > > > - The student and the teacher are both identical models which consist of an encoder and decoder, as depicted in Fig. 1(c).
> > > > > > > - Therefore the analysis indeed applies to the decoder because both the encoder and decoder are receiving gradients from the consistency loss.
> > > > > > >
> > > > > > > ---
> > > > > > >
> > > > > > > > *And so, then one should expect to see the similar order in Figs. 2-3(b)? (Note again that I do not understand where the order you mention comes from.)*
> > > > > > >
> > > > > > > - The reconstruction and consistency loss signals are independent objectives, and the upper bound in Eq. 5 only applies to the consistency loss. So no, we have no reason to expect to see a similar order of gradient norms in Figs. 2-3(b).
> > > > > > > - Again, the order in Figs. 2-3(a) comes from interaction of the dot product $\mathbf{\hat{x}}^\top \mathbf{\tilde{x}}$ in the gradient norm in Eq. 5, where the RHS gives the upper bound.
> > > > > > > - Our analysis only predicts behaviors of the **consistency** loss gradient, which we have also observed in Figs. 2-3.
> > > > > > >
> > > > > > > ---
> > > > > > >
> > > > > > > ### Thank you for the continued discussion, we sincerely hope we have clarified some misunderstandings and we remain open to further discussion during the rest of the discussion period.

---

### Official Review · Reviewer_JA1p · 2022-10-21

**Confidence:** 2
**Correctness:** 3
**Technical Novelty And Significance:** 3
**Empirical Novelty And Significance:** 2
**Recommendation:** 6

**Clarity, Quality, Novelty And Reproducibility:**

Clarity: The paper is well written and easy to follow. This paper makes the points clear in the math proofs.

Novelty: The reviewer is confused about RC-MAE being a major contribution or just a validation for EMA teacher dynamics. For this simple method, greater experimental improvement are expected.

Reproducibility: This paper provide sufficient details to reproduce the results.


**Strength And Weaknesses:**

**Strength**:
[1] The theoretical analysis of EMA teacher dynamics is solid.
[2] The reviewer appreciate that the RC-MAE objective does not introduce a new hyper parameter. The reconstruction term and consistency term simply add together without a balance.


**Weakness**:
[1] Although RC-MAE outperforms MAE on classification, detection and segmentation benchmarks, the improvement is not significant enough.
[2] With all the mathematical derivations, perhaps more insights and conclusions could be obtained.

**Summary Of The Paper:**

This paper firstly analyzes the properties of mean teacher in a masked data modeling task with a simplified linear model. The mean teacher contributes to the training by gradient correction, and it is similar to the memory queue. Several empirical findings about the mean teacher are introduced based on the experimental observation.
Next, the paper proposes RC-MAE (Reconstruction-Consistent Masked Auto-Encoder), which add a EMA teacher for MAE. The proposed model achieves better performance compared to the baseline on classification, detection and segmentation benchmarks. Besides, RC-MAE costs less computation resource and obtains more stability.

**Summary Of The Review:**

This paper explains the role of mean teacher in a MIM model and proposes RC-MAE that achieves better performance than MAE. With good quality and fair technical contributions, this paper is marginally above the acceptance threshold.

---

> ### Author Response · Authors · 2022-11-08
> **Author Response**
>
> Thank you for your valuable comments and appreciation for our work. Below we discuss the points you have raised in detail.
>
> >*Although RC-MAE outperforms MAE on classification, detection and segmentation benchmarks, the improvement is not significant enough.*
>
> - We would like to stress that the main point of our work is to investigate the interactions between the student and EMA teacher in SSL, which to our knowledge, has been missing in the SSL literature.
> - We have provided a novel derivation and explanation of the mechanism by which the teacher helps during training. We strongly advocate that understanding the interactions between the student and teacher can allow future works to utilize and extend this knowledge.
> - In addition to this, RC-MAE leads to improved generalization,  and we have shown that the simple addition of an EMA teacher can allow significantly **_faster_** convergence (Fig. 1(b), e.g., RC-MAE, 800 epochs, 142.6 hours ~= MAE, 1600 epochs, 256.5 hours)
>   - **Training cost** (AWS price**, $) : 4,450 (RC-MAE) vs. 8,005 (MAE); please refer to the above **Shared response**.
>
> >*With all the mathematical derivations, perhaps more insights and conclusions could be obtained.*
> - In our work, we attempted to explore how EMA teachers interact with student models in MIM. From our analysis, we have made the observations that based on the feature similarity of the current inputs to previous inputs, there is a conditional gradient correction provided by the teacher.
> - We then empirically observed this effect in both simple models (Fig. 2) and the full RC-MAE model, which is a deep ViT (Fig. 3), where we observed that based on the aforementioned feature similarity, the **teacher gives a correspondingly strong gradient signal in a predictable relative direction**.
> - The most striking example of this is in (Figure 3c) where we show that the effect derived in the equations for a linear model is still present in the full RC-MAE ViT model. When the features are the same or similar to previous inputs, the teacher responds with a strong gradient in the opposite direction, while when the inputs are very different, the teacher responds with a smaller gradient in an orthogonal direction. To our knowledge, our work is the **_first_** to propose and observe such behavior between student/teacher models. We look forward to seeing future analyses into the dynamics of the student and teacher, as we feel that understanding the dynamics will ultimately lead to better generalizing SSL models.
>
> >*The reviewer is confused about RC-MAE being a major contribution or just a validation for EMA teacher dynamics. For this simple method, greater experimental improvements are expected.*
>
> - Our main goal is to explore the effects of EMA teachers in modern SSL.
> - Creating RC-MAE allows us to have a chance to compare to a well-known baseline and also observe the effect of adding an EMA teacher.
> - Therefore, RC-MAE is used for both observing the predicted dynamics as well as an overall performance evaluation, but we stress that our main goal lies in explaining the teacher dynamics which have not been adequately explained in previous work.
> - We would like to highlight the fact that the use of EMA teachers is widespread in SSL, but there has been no in-depth analysis like ours that we know of to date. Most prior works provide loose intuitions as to the behavior of the teacher without concretely analyzing how the teacher affects training.
>
>
> Thank you for reading this response. We would be happy to discuss any further questions or concerns!

---

### Official Review · Reviewer_jGgU · 2022-10-23

**Confidence:** 4
**Clarity, Quality, Novelty And Reproducibility:** The idea is not novel and there is no…
**Correctness:** 3
**Technical Novelty And Significance:** 3
**Empirical Novelty And Significance:** Not applicable
**Recommendation:** 5

**Strength And Weaknesses:**

Strength:
+ This paper gives a analysis contributation of EMA teacher in self-supervised learning, and finds that the gradient provided by the teachers conditionally ajusts current gradient direction and magnitude conditioned on the similarity of features.
+ EC-MAE achieves similarly performance for MAE with faster convergence speed, aversatial robustness.
+ EC-MAE saves both memory and computation compared with self-distillation-based MIM methods.


Weaknesses:
+ The performance on ImagNet-1K is not impressive compared with recent concurrent works.
+ Lack of downstream tasks like semantic segmentation tasks
+ Why proposed method achieve better robustness, could you give some explanation?

**Summary Of The Paper:**

This paper proposes a simple SSL method, the Reconstruction-COnsistent Masked Auto-Encoder (RC-MAE) by adding an EMA teacher to MAE. Experiments show that RC-MAE converges faster and requires less memory usage than state-of-the-art self-distillation methods during pre-training. Compared with MAE, RC-MAE consistently outperforms on various tasks, including image classification on ImageNet-1K, object detection and instance segmentation on COCO.

**Summary Of The Review:**

See Strength And Weaknesses

---

> ### Author Response · Authors · 2022-11-08
> **Author Response**
>
> Thank you for your valuable comments and appreciation for our work. Below we discuss the points you have raised in detail.
>
> > *The performance on ImagNet-1K is not impressive compared with recent concurrent works.*
>
> - We must stress that the main purpose of our work lies in the exploration and novel analysis of EMA teachers in SSL methods, and our goal was not to directly beat all SOTA results.
> - For ViT-S and ViT-B, our baseline MAE shows lower performance than the state-of-the-art self-distillation methods, thus as an extension of MAE, RC-MAE improves over MAE, but is still lower than some baselines in Table 4. However, we would like to emphasize that **efficiency** is as important as performance. Our RC-MAE allows for **faster convergence** (Fig. 1(b)) than MAE and requires **lower memory overhead & computation cost** (Table 6.) compared to the state-of-the-art self-distillation methods while achieving better ImageNet generalization for **ViT-L**.
>   - **Training cost** (AWS price**, $) : 4,450 (RC-MAE) vs. 8,005 (MAE); please refer to the above **Shared response**.
> - We would like to note that some concurrent works such as BEiTv2 use external pretrained modules (e.g., CLIP). Additionally, the main objective of our work is to **investigate** the role of EMA teacher theoretically and empirically, therefore we design a simple extension on top of MAE by excluding any other architecture tuning e.g., decoder depth/dimension. Thus, it can be expected that adding more sophisticated modules to our RC-MAE yields synergy.
>
>
> | Method            | Finetuning top-1 acc. | Memory | Throughput |
> |-------------------|:----------:|:------:|:----------:|
> | MSN [1]              |      -     |  183GB |  78 imgs/s |
> | iBOT [2]             |    84.8    |  227GB | 123 imgs/s |
> | BootMAE [3]          |    85.9    |  98GB  | 376 imgs/s |
> | **RC-MAE (ours)** |    **86.1**    |  **95GB**  | **441 imgs/s** |
> - Since MSN does not provide ViT-L accuracy, we measure memory consumption and GPU time by using their official code. More details for measurement are described in Appendix F.2.
>
>
> >*Lack of downstream tasks like semantic segmentation tasks*
>
> - Except for ImageNet-1K and COCO detection/instance segmentation, MAE does not provide any code or hyper-parameter information (e.g., learning rate, optimizer, etc). Thus,  for a fair comparison with MAE,  we evaluated on ImageNet-1K and COCO. Furthermore, compared to ImageNet-1K (1M) and COCO (123,288),  as a semantic segmentation dataset, ADE20K(22,210) is a much smaller dataset, which leads to overfitting and thus requires hyper-parameter search. Therefore, we focused on ImageNet-1K and COCO to validate our method on top of MAE.
>
> > *Why proposed method achieve better robustness, could you give some explanation?*
> - It is important to note that the increased robustness is a direct result of adding an EMA teacher, as there is otherwise no difference between the two models in Table 7. Our hypothesis is that the conditional gradient correction observed in a ViT (Fig. 3) suggests that the gradient signal from redundant features is given less weight which should allow the model to move toward a solution based on the *newer* knowledge gained through seeing the current batch while giving less weight to redundant knowledge. Thanks to the **adaptiveness** to the *new* signal, the pre-trained RC-MAE copes well with ImageNet-Corruption or Perturbation which adds noise (e.g., Gaussian) in a pixel-wise manner.
>
> >*The idea is not novel and there is no code for reproduction.*
> - We would also like to point out that the main novelty of our work is to **analyze** how EMA teachers aid the student models during training, something which has been largely _missing_ from prior works regarding EMA teachers. To our knowledge, our work is the **_first_** to identify and observe such behavior between student/teacher models.
> - This analysis and understanding will be important for all SSL methods which utilize an EMA teacher, as it may lead to future researchers designing more efficient or effective teachers.
> - We are not sure why the claim of no code is made, as we have included the code in the original zipped supplementary file, and plan on publicly releasing the code on GitHub as well upon acceptance. Is there something specific from the supplementary file which is missing?
>
> Thank you for reading this response. We would be happy to discuss any further questions or concerns!
>
>
> ----
> [1] *Assran et al. "Masked siamese networks for label-efficient learning." In ECCV, 2022.*
> [2] *Zhou et al. "ibot: Image bert pre-training with online tokenizer." In ICLR, 2022.*
> [3] *Dong et al. "Bootstrapped Masked Autoencoders for Vision BERT Pretraining." In ECCV, 2022.*

---

### Official Review · Reviewer_rLZG · 2022-10-24

**Confidence:** 3
**Correctness:** 3
**Technical Novelty And Significance:** 2
**Empirical Novelty And Significance:** 2
**Recommendation:** 6

**Clarity, Quality, Novelty And Reproducibility:**

The originality is OK. The writing is clear and easy to follow. The overall quality is good.

**Strength And Weaknesses:**

Strengths

- simple and effective
- proved mean teachers technique is compatible with ViT-based MAE empirically
- slightly stronger representation and much faster self-supervised training but requires not very much extra memory/computation cost compared to MAE, making RC-MAE more scalable compared to MAE and more practical than other self-distillation MIM methods (MSN, iBOT)

Weaknesses

- The final solution is exactly the combination of two well-known techniques, which makes sense but is too straightforward. Also, introducing some kind of memory mechanism to improve MAE has been demonstrated effective in BootMAE, which proposed a momentum encoder using EMA update. I think in-depth analysis and a thorough comparison with BootMAE are required.
- The accompanied mathematical analysis and the synthetic experiments are good. But it is an extremely simplification of the actual model (ViT), which makes me wonder if the conclusions drawn from the equations are applicable to RC-MAE.

**Summary Of The Paper:**

This submission proposed a new simple yet effective method, named Reconstruction-Consistent Masked Auto-Encoder (RC-MAE), by equipping the latest ViT-based Masked image modeling (MIM) with mean teachers.
The authors derive some approximations (using a simple linear model) on MIM pretext setting to analyze the role of mean teachers. Some synthetic experiments are provided to support this mathematical analysis. Finally, experiments on classification (ImageNet-1k), and object detection (COCO) tasks show consistent improvement over MAE and with faster training time (RC-MAE, 800 epochs, 166.6 hours ~= MAE, 1600 epochs, 256.5 hours).

**Summary Of The Review:**

Although the proposed method is effective, it is very straightforward and provides very much ``expected'' results. The analysis and conclusion from an extremely simplified linear model may not be the same as the actual ViT-based model.

---

> ### Author Response · Authors · 2022-11-08
> **Author Response**
>
> Thank you for your valuable comments and appreciation for our work. Below we discuss the points you have raised in detail.
>
> > *The accompanied mathematical analysis and the synthetic experiments are good. But it is an extremely simplification of the actual model (ViT), which makes me wonder if the conclusions drawn from the equations are applicable to RC-MAE.*
>
> - Indeed, the linear model is an extreme simplification of a ViT. We used the linear model for the derivation to gain the insight that the consistency gradient magnitude and direction should depend on feature similarities.
> - We then ran empirical tests on the linear model (Fig. 2) to observe if feature similarities really do affect the size and direction of the gradients from the teacher, which confirmed our hypothesis in the linear model.
> - We then **applied** the same empirical test to the **ViT model (Fig. 3)** and observed that the predicted trend from the linear model still holds in the full RC-MAE model (which is a ViT). Fig. 3 shows that the size (Fig. 3(a)) and direction (Fig. 3(c)) of  of the consistency gradient depend on the feature similarity.
> - Therefore, yes, the conclusions drawn from the equations in the linear model have been empirically observed in RC-MAE, a ViT.
> - The figure titles stated the model was RC-MAE, but we have updated to caption to clarify that Fig. 3 shows the full RC-MAE which is in fact a deep ViT.
>
> > ### **Simplicity**
>
> - Since our main goal of this work is to investigate the dynamics of EMA teacher in SSL literature with a **_minimum_** modification, we designed RC-MAE,  trying to exclude other factors (e.g., additional modules) besides the addition of an EMA teacher.
> - Aside from the performance gain, our method can significantly reduce training time, which can be exhaustive in previously proposed SSL methods. (RC-MAE, 800 epochs, 142.6 hours ~= MAE, 1600 epochs, 256.5 hours)
>   - **Training cost** (AWS price**, $) : 4,450 (RC-MAE) vs. 8,005 (MAE); refer to **Shared response**.
> - Despite the simplicity, RC-MAE with **ViT-L** achieves higher accuracy, while requiring less computation and memory consumption compared to the state-of-the-art self-distillation methods (MSN[1], iBOT[2], and BootMAE[3]) with more complex modules as shown in the table below.
>
> | Method            | Finetuning top-1 acc. | Memory | Throughput |
> |-------------------|:----------:|:------:|:----------:|
> | MSN [1]              |      -     |  183GB |  78 imgs/s |
> | iBOT [2]             |    84.8    |  227GB | 123 imgs/s |
> | BootMAE [3]           |    85.9    |  98GB  | 376 imgs/s |
> | **RC-MAE (ours)** |    **86.1**    |  **95GB**  | **441 imgs/s** |
>
>
> > ### **Comparison with BootMAE** [3] in the perspective of student \& teacher consistency.
> - **Different target** from EMA teacher: feature vs. pixel.
>   - BootMAE also uses an EMA teacher by adding a feature-level prediction task on top of a masked auto-encoder architecture. While BootMAE adopts the EMA teacher to provide **feature**-level target to the student, we allow the EMA teacher to reconstruct the masked patches in the same way as the student, providing a **pixel**-level target for reconstruction-consistency.
> - **Asymmetric vs. Symmetric**.
>   - In BootMAE, the student network is equipped with a pixel regressor and a feature predictor for the decoder blocks while the EMA teacher only has an encoder. However, unlike BootMAE, our RC-MAE has a symmetric structure, using the same architecture for student and EMA teacher. Due to the more complex decoders and modules in BootMAE’s student network, it incurs a higher memory and computational cost compared to RC-MAE.
> - **Full patches vs. unmasked patches**.
>   - BootMAE gives inputs to the student and EMA teacher differently. Specifically, BootMAE gives the full set of patches (i.e., full image) to the EMA teacher, while the student is only given unmasked patches similar to MAE. However, RC-MAE only gives unmasked patches to both the EMA teacher and the student, which results in a reduction of memory usage and computation.
> - **Scalability**. Using a bigger backbone (e.g., ViT-L) tends to overfit. For example,
> for ViT-B, BootMAE shows higher performance than RC-MAE, but for ViT-L, the result is the opposite. As BootMAE has more complex decoders and modules, it increases model size when using the bigger model, e.g., ViT-L.
> This suggests two important advantages over BootMAE as below:
>   1) Its simplicity allows RC-MAE to reduce the burden when scaling the model.
>   2) the regularization provided by RC-MAE prevents overfitting in larger models more effectively.
>
>
> Thank you for reading this response. We would be happy to discuss any further questions or concerns!
>
> ----
> [1] *Assran et al. "Masked siamese networks for label-efficient learning." In ECCV, 2022.*
> [2] *Zhou et al. "ibot: Image bert pre-training with online tokenizer." In ICLR, 2022.*
> [3] *Dong et al. "Bootstrapped Masked Autoencoders for Vision BERT Pretraining." In ECCV, 2022.*

---

### Author Response · Authors · 2022-11-08
**Shared Response**

We would like to say thank you to the reviewers for taking the time to evaluate our work. In addition to individualized responses, we would like to bring attention to a few important points in this general response.

## On Novelty

- A common theme across reviews regards the analysis as novel/solid/good, but also raises concerns over novelty of the model. We would like to stress that the **analysis** of the student/EMA-teacher paradigm is the **main** intent of our work, and the RC-MAE model was proposed as an extension to a simple and well-known method (MAE) so we could attempt to observe the effects of adding an EMA teacher.
- Previous uses of EMA teachers only provide loose intuitions as to the mechanism/information provided by the teacher, WITHOUT properly explaining what the teacher provides. ***We aimed to identify and explore what the teacher brings in more detail.***
  - Acting as a *conditional momentum regularizer*
- We emphasize a **distinct difference** from the recent self-distillation methods in the perspective of how to use EMA teacher.
  - MSN [1], iBOT [2], and BootMAE [3] feed the full image patches into EMA teacher, while RC-MAE passes the only visible patches (e.g., w/o mask) to EMA teacher as the student, which leads to reducing computation and memory cost.
- **Understanding** EMA teachers is important and necessary for future works, as any insights may inspire other researchers to design more efficient and effective teacher models, and further analyze the role of EMA teachers.
- Therefore, we believe our contribution is meaningful for all current and future works which utilize EMA teachers, and we expect that our analysis of the dynamics of the EMA teacher will be helpful to the SSL community. Furthermore, we hope RC-MAE is can be easily adopted thanks to its simplicity.

## On Marginal Gains and Efficiency

- Given that our goal centered on the analysis of the EMA teacher, we did not set out to beat all SOTA results with a new model. We aimed instead to take a well-known, current SSL/ViT-based model, and attempt to observe the effects predicted by our analysis. **We observed the effects in both a linear model (Fig. 2) and the ViT based MAE (Fig. 3)**, with additional overall performance evaluations as well.
- We found the simple addition of a teacher is competitive with SOTA results, and improves convergence speed and performance over the base MAE student model. As the only difference between RC-MAE and the original MAE is the teacher, these effects must stem directly from the added **gradient correction** provided by the teacher which is highlighted in our analysis.
- We designed RC-MAE to be a **_minimal_** modification of the base MAE model by excluding other factors (e.g., complex or external-pretrained modules as BEiT) besides the addition of an EMA teacher, which ensures the benefit of EMA teacher is well-isolated.
 - Aside from the performance gain, our method can significantly reduce **training time**, which can be exhaustive in previously proposed SSL methods. (RC-MAE, 800 epochs, 142.6 hours ~= MAE, 1600 epochs, 256.5 hours)
   - **Training cost** (AWS price**, US\$) : 4,450 (RC-MAE) vs. 8,005 (MAE)
- Despite the simplicity, RC-MAE with **ViT-L** achieves higher accuracy, while requiring less **computation** and **memory consumption** compared to the state-of-the-art self-distillation methods (MSN[1], iBOT[2], and BootMAE[3]) with more complex modules as shown in the table below.

| Method            | Finetuning top-1 acc. | Memory | Throughput |
|-------------------|:----------:|:------:|:----------:|
| MSN [1]              |      -     |  183GB |  78 imgs/s |
| iBOT [2]             |    84.8    |  227GB | 123 imgs/s |
| BootMAE [3]          |    85.9    |  98GB  | 376 imgs/s |
| **RC-MAE (ours)** |    **86.1**    |  **95GB**  | **441 imgs/s** |
- Since MSN does not provide ViT-L accuracy, we measure memory consumption and GPU time by using their official code. More details for measurement are described in Appendix F.2.

----
[1] *Assran et al. "Masked siamese networks for label-efficient learning." In ECCV, 2022.*
[2] *Zhou et al. "ibot: Image bert pre-training with online tokenizer." In ICLR, 2022.*
[3] *Dong et al. "Bootstrapped Masked Autoencoders for Vision BERT Pretraining." In ECCV, 2022.*

** We estimate the GPU cost ($) by using AWS service, referring to the below links.
  - we used  8x NVIDIA V100 GPUs with 32 GiB.
  - $31.212 per hour when using the `p3dn.24xlarge` equipped with 8x V100 GPUs in AWS.
- https://instances.vantage.sh/aws/ec2/p3dn.24xlarge
- https://aws.amazon.com/ec2/pricing/on-demand/?nc1=h_ls

---

### Author Response · Authors · 2022-11-17
**A Kindly Reminder Message**

Dear Reviewers,

We appreciate your time and effort in reviewing our work. We have done our best to address each point raised during the discussion period. If you still have any remaining questions or concerns, we would sincerely like to know and will try our best to resolve them within the remaining discussion period.

Thank you,
Authors

---

### Author Response · Authors · 2022-11-25
**Have We Addressed Your Comments?**

### Dear Reviewers,

We sincerely hope we have adequately addressed each comment, and we remain open to further discussion until the end of the discussion period. Please see our brief summary below:

---

### jGgU, JA1p, rLZG

In our response to the initial review, we have highlighted the following:

- Our main goal in this work was to analyze the behavior of EMA teacher models to further understand **_how the teacher helps_** the learning process.
  - Acting as a _conditional momentum regularizer_.
- We feel this analysis provides crucial insights to the research community, where this popular method has seen wide use, without adequate analysis and understanding.
- In addition to identifying and observing characteristics of the training dynamics in both linear and ViT models, we proposed a simple model which is competitive with state-of-the-art methods and achieves faster convergence, low memory/computation overhead and decreased training costs.

---

### 5RHA

In our initial and continued discussion, we have highlighted the following:

- The drop in the gradient norms seen Fig. 3(a) is caused by the decaying learning rate in the MAE training routine.
- Additionally, the behavior in Figs. 2-3 (linear and ViT model, respectively) which is explained by our analysis is:
  1. The order of the lines in Fig. 2-3(a)
  2. The directions of the resulting reconstruction and consistency gradients being commensurately opposite.
- This behavior does occur for random batches of samples.

---

### We value the time and effort in reviewing our work. We remain open to further discussion, and we sincerely hope we have addressed all of your comments.

### Thank you,
### Authors

---

### Decision · Program_Chairs · 2023-01-20

**Decision:**

Accept: poster

**Justification For Why Not Higher Score:**

The novelty of the method is limited, and the main analysis in this paper is for a linear model.  While the paper is valuable, these two factors make it not as interesting to be a spotlight or oral.

**Justification For Why Not Lower Score:**

The review provided by Reviewer jGgU is too short to draw conclusion. Reviewer 5RHA's concerns were addressed by the reviewers in my view (although over email when I reached out, the reviewer still insisted about their claims).

**Metareview: Summary, Strengths And Weaknesses:**

*Summary* The paper proposes a self-supervised learning method for images. The method, called RC-MAE, combines Masked Image Modeling with a mean teacher. The paper analyzes the role of these mean teachers for linear models, and shows some analysis on non-linear (ViT) models. The experiments are conducted on the ImageNet dataset.

*Strengths*: (1) The analysis in the paper is quite interesting. Mean teachers are used for representation learning but aren't well understood. The paper's analysis on mean teachers, their role in the gradient updates is likely to be useful for future work. (2) The method that uses both reconstruction (pixel) and feature matching loss shows empirical gains on ImageNet and other benchmarks. (3) The method is efficient and provides good performance without a lot of training. This makes it useful in practice.

*Weaknesses*: (1) The analysis for a simple linear model does not hold for a complex non-linear model used in practice. The only evidence for this is empirical (Figure 3). (2) The final method is a combination of prior work with similarity to other recent work like BootMAE.

**Note From Pc:**

if the above contains the word "oral" or "spotlight" please see: "oral" presentation means -> notable-top-5% and "spotlight" means -> notable-top-25%. As stated in our emails, we are disassociating presentation type from AC recommendations